# Bounded-Regret MPC via Perturbation Analysis: Prediction Error, Constraints, and Nonlinearity

**Yiheng Lin**
California Institute of Technology
Pasadena, CA, USA
yihengl@caltech.edu

**Yang Hu**
Harvard University
Cambridge, MA, USA
yanghu@g.harvard.edu

**Guannan Qu**
Carnegie Mellon University
Pittsburgh, PA, USA
gqu@andrew.cmu.edu

**Tongxin Li**
The Chinese University of Hong Kong (Shenzhen)
Shenzhen, Guangdong, China
litongxin@cuhk.edu.cn

**Adam Wierman**
California Institute of Technology
Pasadena, CA, USA
adamw@caltech.edu

## Abstract

We study Model Predictive Control (MPC) and propose a general analysis pipeline to bound its dynamic regret. The pipeline first requires deriving a perturbation bound for a finite-time optimal control problem. Then, the perturbation bound is used to bound the per-step error of MPC, which leads to a bound on the dynamic regret. Thus, our pipeline reduces the study of MPC to the well-studied problem of perturbation analysis, enabling the derivation of regret bounds of MPC under a variety of settings. To demonstrate the power of our pipeline, we use it to generalize existing regret bounds on MPC in linear time-varying (LTV) systems to incorporate prediction errors on costs, dynamics, and disturbances. Further, our pipeline leads to regret bounds on MPC in systems with nonlinear dynamics and constraints.

## 1  Introduction

Model Predictive Control (MPC) is an optimal control approach that solves a Finite-Time Optimal Control Problem (FTOCP) using future predictions in a receding horizon manner [1]. It is a flexible approach that is able to accommodate nonlinear and time-varying dynamics, state and actuation constraints, and general cost functions [2–5]. As a result, it is broadly applied in a wide spectrum of control problems, including robotics [6–10], autonomous vehicles [11–17], power systems [18–24], process control [25–27], etc.

Despite the popularity of MPC, its theoretic analysis has been quite challenging. Early works along this line focused on the stability and recursive feasibility of MPC [28–31]. More recently, there has been tremendous interest in providing finite-time learning-theoretic performance guarantees for MPC, such as regret and/or competitive ratio bounds [32, 33]. For example, progress has recently been

---

†This work is supported by NSF Grants CNS-2146814, CPS-2136197, CNS-2106403, NGSDI-2105648, EPCN-2154171, with additional support from Amazon AWS. Yiheng Lin was supported by Kortschak Scholars program. Tongxin Li was supported by the start-up funding UDF01002773 of CUHK-Shenzhen.

made toward (i) regret analysis of MPC in linear time-invariant (LTI) systems with prediction errors on the trajectory to track [34], (ii) the dynamic regret and competitive ratio bounds of MPC under linear time-varying (LTV) dynamics with exact predictions [35], and (iii) exponentially decaying perturbation bounds of the finite-time optimal control problem in time-varying, constrained, and non-linear systems [36, 37]. Beyond MPC, providing regret and/or competitive ratio guarantees for a variety of (predictive) control policies has been a focus in recent years. Examples include RHGC [38, 39] and AFHC [20, 40] for online control/optimization with prediction horizons, OCO-based controllers [41, 42] for no-regret online control, and variations of ROBD for competitive online control without predictions [43, 44] or with delayed observations [45]. In addition, regret lower bounds have been studied in known LTI systems [46] and unknown LTV systems [47].

A promising analysis approach that has emerged from the literature studying MPC and, more generally, predictive control, is the use of perturbation analysis techniques, or more particularly, the use of so-called exponential decaying perturbation bounds. Such techniques underlie the results in [34–37]. This research direction is particularly promising since perturbation bounds exist for FTOCP in many dynamical systems, e.g., [48–52], and thus it potentially allows the derivation of regret and/or competitive ratio bounds in a variety of settings. However, to this point the approach has only yielded results in unconstrained linear systems with no prediction errors (e.g., [35]), and often requires adjusting MPC to include a counter-intuitively large re-planning window due to technical challenges in the analysis (e.g., [48, 49]).

Thus, though perturbation analysis techniques might seem promising, many important questions about applying them for the study of predictive control remain open. Firstly, one of the major reasons for the extensive application of MPC is its flexibility in incorporating constraints and nonlinear dynamics [53]. However, none of the existing results and approaches can analyze the performance of MPC under constraints and/or nonlinear dynamics. In fact, the anlyasis of MPC under constraints or nonlinearity has long been known to be challenging because of the intractable form of cost-to-go functions and optimal solutions. Secondly, prediction error is inevitable for real-world implementations of MPC due to unpredictable noise and model mismatch, yet the analysis of MPC subject to prediction errors is limited. Thirdly, existing approaches analyze MPC in a case-by-case manner and, in most cases, the analysis framework is specific to the assumptions of the particular case (e.g. quadratic costs, perfect predictions, etc) in a way that does not generalize to other settings [33–35, 48, 49].

**Contributions.** In this paper, we propose a general analysis pipeline (Section 3) that converts perturbation bounds for an FTOCP into dynamic regret bounds for MPC across a variety of settings. More specifically, the pipeline consists of three steps (see Figure 1). In Step 1, we obtain the required perturbation bounds for the specific setting. In Step 2, as shown in Lemma 3.1, the perturbation bounds are used to bound the *per-step error*, which is defined to be the error of the MPC action against the clairvoyant optimal action (see Definition 3.1). In Step 3, the per-step error bound is converted to a dynamic regret bound for MPC, as shown in Lemma 3.2. The full pipeline is summarized into a *Pipeline Theorm* (Theorem 3.3), which directly converts perturbation bounds into bounds on the dynamic regret of MPC in general settings, including those with time-variation, prediction error, constraints, and nonlinearities. The key technical insight that enables the pipeline is the following recursive relationship between Step 2 and Step 3 (Lemma 3.1 and Lemma 3.2): Step 2 guarantees a "small" per-step error $e_t$ once the current state $x_t$ of MPC is "near" the offline optimal trajectory (OPT), while Step 3 guarantees the next state $x_{t+1}$ of MPC will be near OPT if all previous per-step errors ($\{e_\tau\}_{\tau \leq t}$) are small. Thus Step 2 and Step 3 work together to guarantee MPC states are always near OPT and thus MPC per-step errors are always small (Theorem 3.3).

To demonstrate the power of the proposed pipeline, we apply it to a range of settings, as summarized in Table 1. Our first applications are to two settings with linear time-varying (LTV) dynamics and prediction errors on (i) disturbances, Section 4.1, and (ii) the dynamical matrices and cost functions, Section 4.2. The state-of-the-art results in the LTV setting are [35], which requires exact knowledge of the disturbances and of the dynamics. To the best of our knowledge, our work provides the first regret result for MPC with prediction error on the dynamics (see Theorem 4.2), a result that enables the bounds in settings where MPC is applied to learned dynamics [54].

Our second application is to a setting with nonlinear dynamics and constraints (Section 5). We show the first dynamic regret bound for MPC under state and actuation constraints in nonlinear systems with general costs (Theorem 5.1). Very few prior results exist for MPC in this setting, even with nonlinear dynamics or constraints individually. The most related works are [48], which studies constrained

MPC, and [49], which studies nonlinear MPC. In both cases, a counter-intuitive re-planning window is added to MPC to facilitate the analysis, a downside that our pipeline could avoid. Besides, [48] and [49] require exact predictions of the cost functions, dynamics, and constraints for the exponential convergence property of MPC to hold, while our result can apply to more general noisy predictions.

## 2 Preliminaries

In this section, we first introduce the general predictive online control problem including the settings, the objective, available information, and the predictive controller class. Then, we introduce the MPC algorithm, which is a widely-used predictive controller that we focus on in this work. Specifically, we consider a general, finite-horizon, discrete-time optimal control problem with *time-varying costs, dynamics and constraints*, namely

$$\min_{x_{0:T}, u_{0:T-1}} \sum_{t=0}^{T-1} f_t(x_t, u_t; \xi_t^*) + F_T(x_T; \xi_T^*)$$

$$\text{s.t. } x_{t+1} = g_t(x_t, u_t; \xi_t^*), \qquad\qquad \forall 0 \le t < T,$$
$$s_t(x_t, u_t; \xi_t^*) \le 0, \qquad\qquad \forall 0 \le t < T, \qquad (1)$$
$$x_0 = x(0).$$

Here, $x_t \in \mathbb{R}^n$ is the *state*, $u_t \in \mathbb{R}^m$ is the *control input* or *action*; $f_t$ is a time-varying *stage cost* function, $g_t$ is a time-varying *dynamical* function, and $s_t$ is a time-varying *constraint* function, all parameterized by a ground-truth parameter $\xi_t^*$ (unknown to an online controller); and $F_T$ is a terminal cost function parameterized by $\xi_T^*$ that regularizes the terminal state.

The offline optimal trajectory OPT is obtained by solving (1) with the full knowledge of the true parameters $\xi_{0:T}^*$. In contrast, an online controller can only observe noisy estimations of the parameters in a fixed prediction horizon to decide its current action $u_t$ at each time step $t$. For example, MPC picks $u_t$ by calculating the optimal sub-trajectory confined to the prediction horizon. The objective is to design an online controller that can compete against the offline optimal trajectory OPT. We use *dynamic regret* as the performance metric, which is widely used to evaluate the performance of online controllers/algorithms in the literature of online control [32, 34, 35] and online optimization [38, 43, 55]. Specifically, for a concrete problem instance $(x(0), \xi_{0:T}^*)$, let cost(OPT) denote the total cost incurred by OPT, and cost(ALG) denote the total cost incurred by an online controller ALG. The *dynamic regret* is defined as the worst-case additional cost incurred by ALG against OPT, i.e., $\sup_{x(0), \xi_{0:T}^*} (\text{cost}(\text{ALG}) - \text{cost}(\text{OPT}))$.

The formulation in (1) is general enough to include a variety of challenging settings. In this paper, we consider three important settings to illustrate how to apply our analysis pipeline. The settings differ in (a) the form of costs, dynamics, and constraints, and (b) the quantities in the system to be predicted (i.e., parameterized by $\xi_t^*$), and the prediction error allowed. An overview of the settings is presented in Table 1 below.

Table 1: Overview of the settings considered in this paper

| Section | Costs | Dynamics | Constraints | Prediction $\xi_t$ | Prediction error |
|---|---|---|---|---|---|
| 4.1 | decomposable | LTV | none | disturbance: $w_t$ | arbitrary |
| 4.2 | quadratic | LTV | none | cost: $Q_t, R_t, \bar{x}_t$ 
 dynamics: $A_t, B_t$ | sufficiently small |
| 5 | general | non-linear time-varying | non-linear stage constraint | cost: $f_t$ 
 dynamics: $g_t$ 
 constraints: $s_t$ | sufficiently small |

In each setting, we impose different assumptions on cost functions, dynamical systems, constraints, and properties of the predicted quantities as functions of parameter $\xi_t$. In general, we require well-defined costs, Lipschitz and uniformly controllable dynamics, and Lipschitzness of the predicted quantities with regard to $\xi_t$. For constraints, additional assumptions characterizing the active constraints along and near the optimal trajectory are imposed. Detailed definitions and statements are deferred to Appendix B and Sections 3, 4, and 5. To facilitate the statement of the pipeline, we assume the following *universal properties* hold throughout the paper:

- *Stability of* OPT*:* there exists a constant $D_{x^*}$ such that $\|x_t^*\| \le D_{x^*}$ for every state $x_t^*$ on the offline optimal trajectory OPT.

- *Lipschitz dynamics:* the ground-truth dynamical function $g_t(\cdot, \cdot; \xi_t^*)$ is Lipschitz in action; i.e., for any feasible $x_t, u_t, u_t'$, $g_t$ satisfies $\|g_t(x_t, u_t; \xi_t^*) - g_t(x_t, u_t'; \xi_t^*)\| \leq L_g \|u_t - u_t'\|$.

- *Well-conditioned costs:* every stage cost $f_t(\cdot, \cdot; \xi_t^*)$ and the terminal cost $F_T(\cdot; \xi_T^*)$ are nonnegative, convex, and $\ell$-smooth in $(x_t, u_t)$ and $x_T$, respectively.

## 2.1 Predictive Online Control

While Step 3 (Lemma 3.2) in our pipeline can be generally applied to all online controllers, in the subsequent applications we focus on *Model Predictive Control (MPC)*, a popular classical controller. In this subsection, we first define the available information (predictions) as well as its quality (prediction power), and how general predictive online controllers make decisions. Then, we define a useful optimization problem called FTOCP, and introduce MPC as a predictive online controller.

We represent the uncertainties in cost functions, dynamics, constraints, and terminal costs as function families parameterized by $\xi_t$: $\mathcal{F}_t := \{f_t(x_t, u_t; \xi_t) \mid \xi_t \in \Xi_t\}, \mathcal{G}_t := \{g_t(x_t, u_t; \xi_t) \mid \xi_t \in \Xi_t\},$ $\mathcal{S}_t := \{s_t(x_t, u_t; \xi_t) \mid \xi_t \in \Xi_t\}$, and $\mathcal{F}_T := \{F_T(x_T; \xi_T) \mid \xi_T \in \Xi_T\}$. The online controller knows the function families $\mathcal{F}_{0:T}$, $\mathcal{G}_{0:T-1}$, and $\mathcal{S}_{0:T-1}$ as prior knowledge, but it does not know the true parameters $\xi_{0:T}^* \in \prod_{\tau=0}^{T} \Xi_\tau$. Instead, at time step $t$, the online controller has access to noisy predictions of these parameters for the future $k$ time steps (where $k$ is called the *prediction horizon*), represented by $\xi_{t:t+k|t} \in \prod_{\tau=t}^{t+k} \Xi_\tau$. The parameter space $\Xi_t$ at each time step $t$ may have different dimensions.

We formally define the quality of predictions by introducing the following notion of prediction error.

**Definition 2.1.** *The prediction error is defined as* $\rho_{t,\tau} := \left\| \xi_{t+\tau|t} - \xi_{t+\tau}^* \right\|$ *for an integer* $\tau \geq 0$. *The power of $\tau$-step-away predictions (for parameter $\xi$) is defined as* $P(\tau) := \sum_{t=0}^{T-\tau} \rho_{t,\tau}^2$.

Under this noisy prediction model, a general predictive online controller ALG decides the control action based on the current state and the latest available predictions of future parameters. We formally define the class of predictive online controllers considered in this paper in Definition 2.2, which includes MPC as a special case.

**Definition 2.2.** *A predictive online controller* ALG *is a function that takes the current state $x_t$ and the available predictions $\xi_{t:t+k|t}$ as inputs at time $t$ and outputs the current control action $u_t$, i.e.,* $u_t = \mathsf{ALG}(x_t, \xi_{t:t+k|t})$. *We use* $x_0 \xrightarrow{u_0} x_1 \xrightarrow{u_1} \cdots \xrightarrow{u_{T-1}} u_T$ *to denote the trajectory achieved by* ALG, *and use* $x_0 \xrightarrow{u_0^*} x_1^* \xrightarrow{u_1^*} \cdots \xrightarrow{u_{T-1}^*} u_T^*$ *to denote the offline optimal trajectory* OPT.

A core component of both the design of online controllers and our analysis is the following *finite-time optimal control problem* (FTOCP). Given a time interval $[t_1, t_2]$, the FTOCP solves the optimal sub-trajectory subjected to the given initial state $z$, terminal cost $F$, and a sequence of (potentially noisy) parameters $\xi_{t_1:t_2-1}, \zeta_{t_2}$, as formalized in the following definition.

**Definition 2.3.** *The finite-time optimal control problem (FTOCP) over the horizon $[t_1, t_2]$, with initial state $z$, parameters $\xi_{t_1:t_2-1}$ and $\zeta_{t_2}$, and terminal cost $F(\cdot; \cdot)$, is defined as*

$$
\iota_{t_1}^{t_2}(z, \xi_{t_1:t_2-1}, \zeta_{t_2}; F) := \min_{y_{t_1:t_2}, v_{t_1:t_2-1}} \sum_{t=t_1}^{t_2-1} f_t(y_t, v_t; \xi_t) + F(y_{t_2}; \zeta_{t_2})
$$
$$
\begin{aligned}
s.t. \; & y_{t+1} = g_t(y_t, v_t; \xi_t), && \forall t_1 \leq t < t_2, \\
& s_t(y_t, v_t; \xi_t) \leq 0, && \forall t_1 \leq t < t_2, \quad (2) \\
& y_{t_1} = z,
\end{aligned}
$$

*and a corresponding optimal solution as* $\psi_{t_1}^{t_2}(z, \xi_{t_1:t_2-1}, \zeta_{t_2}; F)$. *We shall use the shorthand notation* $\psi_{t_1}^{t_2}(z, \xi_{t_1:t_2}; F) := \psi_{t_1}^{t_2}(z, \xi_{t_1:t_2-1}, \xi_{t_2}; F)$ *when the context is clear.*

Note that the formulation of the FTOCP in Definition 2.3 does not include a terminal constraint set. To compensate for this, we allow the terminal cost $F(\cdot; \zeta_{t_2})$ to take value $+\infty$ in some subset of $\mathbb{R}^n$, and $\zeta_{t_2}$ is not necessarily an element in $\Xi_{t_2}$. For example, a terminal cost function that we frequently use later is the indicator function of the terminal parameter $\zeta_{t_2}$, where $\zeta_{t_2} \in \mathbb{R}^n$. We use $\mathbb{I}$ to denote such indicator terminal cost (i.e., $\mathbb{I}(y_{t_2}; \zeta_{t_2}) = 0$ if $y_{t_2} = \zeta_{t_2}$ and $\mathbb{I}(y_{t_2}; \zeta_{t_2}) = +\infty$ otherwise).

Finally, given the definition of the FTOCP, we are ready to formally introduce MPC. The pseudocode of this online controller is given in Algorithm 1. Basically, at time step $t$, $\mathsf{MPC}_k$ solves a $k$-step predictive FTOCP using the latest available parameter predictions, and commits the first control action in the solution. When there are only fewer than $k$ steps left, $\mathsf{MPC}_k$ directly solves a $(T - t)$-step FTOCP at time $t$ until the end of the horizon, using the predicted real terminal cost $F_T(\cdot; \xi_{T|t})$. This MPC controller (and its variants) has a wide range of real-world applications.

---

**Algorithm 1** Model Predictive Control ($\mathsf{MPC}_k$)

---

**Require:** Specify the terminal costs $F_t$ for $k \leq t < T$.
 1: **for** $t = 0, 1, \ldots, T - 1$ **do**
 2:      $t' \leftarrow \min\{t + k, T\}$
 3:      Observe current state $x_t$ and obtain predictions $\xi_{t:t'|t}$.
 4:      Solve and commit control action $u_t := \psi_t^{t'}(x_t, \xi_{t:t'|t}; F_{t'})_{v_t}$.

---

## 3 The Pipeline: Bounded Regret via Perturbation Analysis

The goal of this section is to give an overview of a novel analysis pipeline that converts a perturbation bound into a bound on the dynamic regret. We begin by highlighting the form of perturbation bounds required in the pipeline, and then describe the 3-step process of applying the pipeline. In subsequent sections, we apply this pipeline to obtain new regret bounds for MPC in different settings.

### 3.1 Per-Step Error and Perturbation Bounds

A key challenge when comparing the performance of an online controller against the offline optimal trajectory is that the online controller's state $x_t$ is different from the offline optimal state $x_t^*$ at time step $t$. Due to such discrepancy in states, we cannot simply evaluate the online controller's action $u_t$ via comparison against the offline optimal action $u_t^*$. To address this challenge, our pipeline uses the notion of per-step error (Definition 3.1) inspired by the performance difference lemma and its proofs in reinforcement learning (RL) [35]. Specifically, we compare $u_t$ to the clairvoyant optimal action one may adopt at the same state $x_t$ if all true future parameters $\xi_{t:T}^*$ are known, which leads to the definition of *per-step error* as follows.

**Definition 3.1.** *The per-step error $e_t$ incurred by a predictive online controller* $\mathsf{ALG}$ *at time step $t$ is defined as the distance between its actual action $u_t$ and the clairvoyant optimal action, i.e.,*

$$e_t := \left\| u_t - \psi_t^T(x_t, \xi_{t:T}^*; F_T)_{v_t} \right\|, \text{ where } u_t = \mathsf{ALG}(x_t, \xi_{t:t+k|t}).$$

*The clairvoyant optimal trajectory starting from $x_t$ is defined as* $x_{t:T|t}^* := \psi_t^T(x_t, \xi_{t:T}^*; F_T)_{y_{t:T}}$.

Note that the clairvoyant optimal trajectory can be viewed as being generated by an MPC controller with long enough prediction horizon and exact predictions. This notion highlights the reason why MPC can compete against the clairvoyant optimal trajectory, since the per-step error in a system controlled by $\mathsf{MPC}_k$ becomes $e_t = \left\| \psi_t^{t+k}(x_t, \xi_{t:t+k|t}; F_{t+k})_{v_t} - \psi_t^T(x_t, \xi_{t:T}^*; F_T)_{v_t} \right\|$. Intuitively, the per-step error converges to zero as the prediction horizon $k$ increases and the quality of predictions improves (i.e. $\left\| \xi_{t:t+k|t} - \xi_{t:t+k}^* \right\| \to 0$).

This intuition highlights the important role of perturbation bounds in comparing online controllers against (offline) clairvoyant optimal trajectories. As we have discussed in Section 1, many previous works [36, 37, 48, 49] have established (local) decaying sensitivity/perturbation bounds for different instances of the FTOCP (2). These bounds may take different forms, but for the application of our pipeline we require two types of perturbation bounds that are both common in the literature:

(a) *Perturbations of the parameters $\xi_{t_1:t_2}$ given a fixed initial state $z$:*

$$\left\| \psi_{t_1}^{t_2}(z, \xi_{t_1:t_2}; F)_{v_{t_1}} - \psi_{t_1}^{t_2}(z, \xi_{t_1:t_2}'; F)_{v_{t_1}} \right\| \leq \left( \sum_{t=t_1}^{t_2} q_1(t - t_1)\delta_t \right) \|z\| + \sum_{t=t_1}^{t_2} q_2(t - t_1)\delta_t,$$

(3)

where $\delta_t := \|\xi_t - \xi_t'\|$ for $t \in [t_1, t_2]$, and scalar functions $q_1$ and $q_2$ satisfy $\lim_{t \to \infty} q_i(t) = 0$, $\sum_{t=0}^\infty q_i(t) \leq C_i$ for constants $C_i \geq 1$, $i = 1, 2$. This perturbation bound is useful in bounding the per-step error $e_t$, as we will discuss in Lemma 3.1.

(b) *Perturbation of the initial state $z$ given fixed parameters $\xi_{t_1:t_2}$*:

$$\left\| \psi_{t_1}^{t_2}\left(z, \xi_{t_1:t_2}; F\right)_{y_t/v_t} - \psi_{t_1}^{t_2}\left(z', \xi_{t_1:t_2}; F\right)_{y_t/v_t} \right\| \le q_3(t - t_1)\left\| z - z' \right\|, \text{ for } t \in [t_1, t_2], \quad (4)$$

where the scalar function $q_3$ satisfies $\sum_{t=0}^{\infty} q_3(t) \le C_3$ for some constant $C_3 \ge 1$. This bound is useful in preventing the accumulation of per-step errors $e_t$ throughout the horizon (see Lemma 3.2). Compared with (3), the right hand side of (4) has a simpler form.

Existing perturbation bounds usually combine the above two types ((3) and (4)) into a single equation that characterizes perturbations on $z$ and $\xi_{t_1:t_2}$ simultaneously, e.g., [35, 37]. Here, we decompose them into two separate types because they are used in different parts of our pipeline.

## 3.2 A 3-Step Pipeline from Perturbation Bounds to Regret

An overview of the pipeline is given in Figure 1, which illustrates the high-level ideas of the pipeline that starts by obtaining perturbation bounds, proceeds to bound the per-step error using perturbation bounds, and finally combines the per-step error and perturbation bounds to bound the dynamic regret. In the following we describe each step in detail.

**Step 1: Obtain the perturbation bounds given in (3) and (4).** The form of the perturbation bounds depends heavily on the specific form of the FTOCP, and thus the derivation requires case-by-case study (e.g., see Section 4 and Section 5). However, off-the-shelf bounds are available in most cases, as there has been a rich literature on perturbation analysis of control systems (e.g., [35–37, 48, 49] and the references therein). The following property summarizes precisely what is expected to be derived for bounds (3) and (4) in Steps 2 and 3.

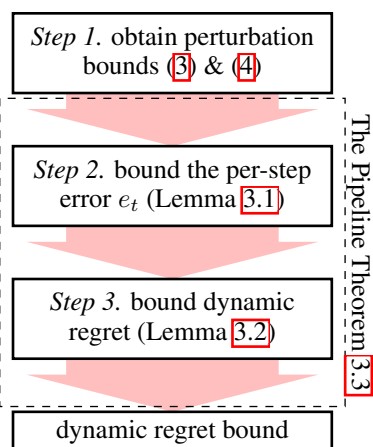

Figure 1: Illustrative diagram of the 3-step pipeline from perturbation analysis to bounded regret.

**Property 3.1.** *Suppose there exists a positive constant $R$ such that the perturbation bound (3) holds for the following specifications: with $t_1 = t$ and $t_2 = t + k$ for $t < T - k$, (3) holds for $F : \mathbb{R}^n \to \mathbb{R}^n$ be the identity function $\mathbb{I}$, and*

$$z \in \mathcal{B}(x_t^*, R); \; \xi_{t:t+k-1} \in \Xi_{t:t+k-1}, \xi'_{t:t+k-1} = \xi_{t:t+k-1}^*; \; \xi_{t+k}, \xi'_{t+k} \in \mathcal{B}(x_{t+k}^*, R) \subseteq \mathbb{R}^n;$$

*with $t_1 = t$ and $t_2 = T$ for $t \ge T - k$, (3) holds for $z \in \mathcal{B}(x_t^*, R)$; $\xi_{t:T} \in \Xi_{t:T}, \xi_{t:T} = \xi_{t:T}^*$; $F = F_T$. Further, perturbation bound (4) holds for any $z, z' \in \mathcal{B}(x_t^*, R)$ and $\xi_{t_1:t_2} = \xi_{t_1:t_2}^*$.*

As a remark, note that for the first specification of Property 3.1 with $t_1 = t$ and $t_2 = t + k$, $\xi_{t+k}$ and $\xi'_{t+k}$ live in the state space $\mathbb{R}^n$ rather than $\Xi_{t+k}$ because they represent the target terminal state of the FTOCP solved by $\mathsf{MPC}_k$. Intuitively, Property 3.1 states that perturbation bounds (3) and (4) hold in a small neighborhood (specifically, a ball with radius $R$) around the offline optimal trajectory OPT, which is much weaker than the global exponentially decaying perturbation bounds required by previous work (e.g., [35]) in the following sense: (i) in the general settings where the dynamical function $g_t$ is non-linear, or where there are constraints on states and actions, one cannot hope the perturbation bound to hold globally for all possible parameters [37, 49, 50]; (ii) the decay functions $\{q_i\}_{i=1,2,3}$ are only required to converge to zero and satisfy $\sum_{\tau=0}^{\infty} q_i(\tau) \le C_i$, which means the exponential decay rate as in [35] is not necessary — in fact, polynomial decay rates can also satisfy these properties, which greatly broadens the applicability of our pipeline.

**Step 2: Bound the per-step error $e_t$.** The core of the analysis is to apply the perturbation bounds to bound the per-step error. For $\mathsf{MPC}_k$, under Property 3.1, this step can be done in a universal way, as summarized in Lemma 3.1 below. A complete proof of Lemma 3.1 can be found in Appendix C.

**Lemma 3.1.** *Let Property 3.1 hold. Suppose the current state $x_t$ satisfies $x_t \in \mathcal{B}(x_t^*, R/C_3)$ and the terminal cost $F_{t+k}$ of $\mathsf{MPC}_k$ is set to be the indicator function of some state $\bar{y}(\xi_{t+k|t})$ that satisfies $\bar{y}(\xi_{t+k|t}) \in \mathcal{B}(x_{t+k}^*, R)$ for $t < T - k$. Then, the per-step error of $\mathsf{MPC}_k$ is bounded by*

$$e_t \le \sum_{\tau=0}^{k}\left(\left(\left(\frac{R}{C_3} + D_{x^*}\right) \cdot q_1(\tau) + q_2(\tau)\right)\rho_{t,\tau}\right) + 2R\left(\left(\frac{R}{C_3} + D_{x^*}\right) \cdot q_1(k) + q_2(k)\right). \quad (5)$$

Lemma 3.1 is a straight-forward implication of perturbation bound (3) specified in Property 3.1. To see this, for $t < T - k$, note that the per-step error $e_t$ can be bounded by

$$e_t = \left\| \psi_t^{t+k}(x_t, \xi_{t:t+k-1|t}, \bar{y}(\xi_{t+k|t}); \mathbb{I})_{v_t} - \psi_t^T(x_t, \xi_{t:T}^*; F_T)_{v_t} \right\| \tag{6a}$$

$$= \left\| \psi_t^{t+k}(x_t, \xi_{t:t+k-1|t}, \bar{y}(\xi_{t+k|t}); \mathbb{I})_{v_t} - \psi_t^{t+k}(x_t, \xi_{t:t+k-1}^*, x_{t+k|t}^*; \mathbb{I})_{v_t} \right\| \tag{6b}$$

$$\leq \sum_{\tau=0}^{k-1} \left( \|x_t\| \cdot q_1(\tau) + q_2(\tau) \right) \rho_{t,\tau} + \left( \|x_t\| \cdot q_1(k) + q_2(k) \right) \left\| \bar{y}(\xi_{t+k|t}) - x_{t+k|t}^* \right\|. \tag{6c}$$

Here, we apply the principle of optimality to conclude that the optimal trajectory from $x_t$ to $x_{t+k|t}^*$ (i.e., $\psi_t^{t+k}(x_t, \xi_{t:t+k-1}^*, x_{t+k|t}^*; \mathbb{I})$ in (6b)) is a sub-trajectory of the clairvoyant optimal trajectory from $x_t$ (i.e., $\psi_t^T(x_t, \xi_{t:T}^*; F_T)$ in (6a)), and (6c) is obtained by directly applying perturbation bound (3). Note that $\|x_t\| \leq \frac{R}{C_3} + D_{x^*}$, and that both $\bar{y}(\xi_{t+k|t})$ and $x_{t+k|t}^*$ are in $\mathcal{B}(x_{t+k}^*; R)$ by assumption and by perturbation bound (4) specified in Property 3.1, we conclude that (5) hold for $t < T - k$. The case $t \geq T - k$ can be shown similarly. We defer the detailed proof to Appendix C.

**Step 3: Bound the dynamic regret by $\sum_{t=0}^{T-1} e_t^2$.** This final step builds upon perturbation bound (4), and aims at deriving dynamic regret bounds in a universal way, as stated in Lemma 3.2 below. Specifically, under the assumption that a local decaying perturbation bound in the form of (4) holds around the offline optimal trajectory OPT, and the property that per-step errors $e_t$ are sufficiently small, we can show that the online controller will not leave the "safe region" near the offline optimal trajectory as specified in Property 3.1, and thus the dynamic regret of ALG is bounded as in (7) (note that ALG is not confined to MPC, but is allowed to be any algorithm with bounded per-step errors). A complete proof of Lemma 3.2 can be found in Appendix D.

**Lemma 3.2.** *Let Property 3.1 hold. If the per-step errors of ALG satisfy $e_\tau \leq R/(C_3^2 L_g)$ for all time steps $\tau < t$, the trajectory of ALG will remain close to OPT at time $t$, i.e. $x_t \in \mathcal{B}(x_t^*, R/C_3)$. Further, if $e_t \leq R/(C_3^2 L_g)$ for all $t < T$, the dynamic regret of ALG is upper bounded by*

$$\text{cost(ALG)} - \text{cost(OPT)} = O\left( \sqrt{\text{cost(OPT)} \cdot \sum_{t=0}^{T-1} e_t^2} + \sum_{t=0}^{T-1} e_t^2 \right). \tag{7}$$

**Summary.** Combining Steps 2 and 3 of the pipeline yields the following *Pipeline Theorem* for $\text{MPC}_k$ (see Theorem 3.3). Basically it states that, when the prediction horizon $k$ is sufficiently large and the prediction errors $\rho_{t,\tau}$ are sufficiently small, Lemma 3.1 and Lemma 3.2 can work together to make sure that $\text{MPC}_k$ never leaves a $(R/C_3)$-ball around the offline optimal trajectory OPT; thus we obtain a dynamic regret bound.

**Theorem 3.3** (The Pipeline Theorem). *Let Property 3.1 hold. Suppose the terminal cost $F_{t+k}$ of $\text{MPC}_k$ is set to be the indicator function of some state $\bar{y}(\xi_{t+k|t})$ that satisfies $\bar{y}(\xi_{t+k|t}) \in \mathcal{B}(x_{t+k}^*, R)$ for all time steps $t < T - k$. Further, suppose the prediction errors $\rho_{t,\tau}$ are sufficiently small and the prediction horizon $k$ is sufficiently large, such that*

$$\sum_{\tau=0}^{k} \left( \left( \frac{R}{C_3} + D_{x^*} \right) \cdot q_1(\tau) + q_2(\tau) \right) \rho_{t,\tau} + 2R \left( \left( \frac{R}{C_3} + D_{x^*} \right) \cdot q_1(k) + q_2(k) \right) \leq \frac{R}{C_3^2 L_g}.$$

*Then, the trajectory of $\text{MPC}_k$ will remain close to OPT, i.e. $x_t \in \mathcal{B}(x_t^*, R/C_3)$ for all time steps $t$, and the dynamic regret of $\text{MPC}_k$ is upper bounded by*

$$\text{cost}(\text{MPC}_k) - \text{cost(OPT)} = O\left( \sqrt{\text{cost(OPT)} \cdot E} + E \right), \tag{8}$$

*where $E := \sum_{\tau=0}^{k-1} (q_1(\tau) + q_2(\tau)) P(\tau) + \left( q_1(k)^2 + q_2(k)^2 \right) T$.*

The proof of Theorem 3.3 can be found in Appendix E. To interpret the dynamic regret bound in (8), note that we have $\text{cost(OPT)} = O(T)$ as a result of our model assumptions. Thus, the dynamic regret of ALG is in the order of $\sqrt{TE} + E$. When there is no prediction error, the regret bound $O((q_1(k) + q_2(k)) \cdot T)$ reproduces the result in [35], and the bound will degrade as the prediction error increases. It is also worth noticing that, when the prediction power improves over time as the online controller learns the system better and $k = \Omega(\ln T)$, the dynamic regret can be $o(T)$.

# 4 Unconstrained LTV Systems

We now illustrate the use of the Pipeline Theorem by applying it in the context of (unconstrained) LTV systems with prediction errors, either on disturbances or the dynamical matrices.

## 4.1 Prediction Errors on Disturbances

In this section, we consider the following special case of problem (1), where the dynamics is LTV and the prediction error can only occur on the disturbances $w_t$:

$$\min_{x_{0:T}, u_{0:T-1}} \sum_{t=0}^{T-1} (f_t^x(x_t) + f_t^u(u_t)) + F_T(x_T)$$
$$\text{s.t. } x_{t+1} = A_t x_t + B_t u_t + w_t(\xi_t^*), \qquad \forall 0 \le t < T, \qquad (9)$$
$$x_0 = x(0).$$

All necessary assumptions on the system are summarized below in Assumption 4.1.

**Assumption 4.1.** *Assume the following holds for the online control problem instance* (9):

- *Cost functions: $\{f_t^x\}_{t=0}^{T-1}, \{f_t^u\}_{t=0}^{T-1}, F_T$ are nonnegative $\mu$-strongly convex and $\ell$-smooth. And we assume $f_t^x(0) = f_t^u(0) = F_T(0) = 0$ without the loss of generality.*
- *Dynamical systems: the LTV system $\{A_t, B_t\}$ is $\sigma$-uniform controllable with controllability index $d$, and $\|A_t\| \le a$, $\|B_t\| \le b$, and $\|B_t^\dagger\| \le b'$ hold for all $t$, where $B_t^\dagger$ denotes the Moore–Penrose inverse of matrix $B_t$.. The detailed definitions can be found in Assumption F.1 in Appendix F.*
- *Predicted quantities: $\|w_t(\xi_t)\| \le D_w$ holds for all $\xi_t \in \Xi_t$ and all $t$. For every time step $t$, $w_t(\xi_t)$ is a $L_w$-Lipschitz function in $\xi_t$, i.e., $\|w_t(\xi_t) - w_t(\xi_t')\| \le L_w \|\xi_t - \xi_t'\|, \forall \xi_t, \xi_t' \in \Xi_t$.*

Under Assumption 4.1, we can again apply the perturbation bounds shown in [35] to show Property 3.1. In particular, we already know that for some constants $H_1 \ge 1$ and $\lambda_1 \in (0, 1)$, perturbation bounds (3) and (4) hold globally for $q_1(t) = 0$, $q_2(t) = H_1 \lambda_1^t$, and $q_3(t) = H_1 \lambda_1^t$. Since both of these perturbation bounds hold globally, radius $R$ in Property 3.1 can be set arbitrarily, and we shall take $R := \max\left\{D_{x^*}, \frac{2L_g H_1^3}{(1-\lambda_1)^3}\right\}$ so that Theorem 3.3 can be applied to $\mathsf{MPC}_k$ with terminal cost $F_{t+k}(\cdot; \xi_{t|t+k}) \equiv \mathbb{I}(\cdot; 0)$. This leads to the following dynamic regret bound:

**Theorem 4.1.** *In the unconstrained LTV setting* (9), *under Assumption 4.1, when the prediction horizon $k$ is sufficiently large such that $k \ge \ln\left(\frac{4H_1^3 L_g}{(1-\lambda_1)^2}\right) / \ln(1/\lambda_1)$, the dynamic regret of $\mathsf{MPC}_k$ (Algorithm 1) with terminal cost $F_{t+k}(\cdot; \xi_{t|t+k}) \equiv \mathbb{I}(\cdot; 0)$ is bounded by $\mathrm{cost}(\mathsf{MPC}_k) - \mathrm{cost}(\mathsf{OPT}) \le O\left(\sqrt{T \cdot \sum_{\tau=0}^{k-1} \lambda_1^\tau P(\tau)} + \lambda_1^{2k} T^2 + \sum_{\tau=0}^{k-1} \lambda_1^\tau P(\tau)\right).$*

A complete proof of Theorem 4.1 can be found in Appendix F. When there are no prediction errors, the bound in Theorem 4.1 reduces to $O(\lambda_1^k T)$, which reproduces the result of [35]. Further, it is also worth noticing that due to the form of discounted sum $\sum_{\tau=0}^{k-1} \lambda_1^\tau P(\tau)$, prediction errors for the near future matter more than those for the far future.

## 4.2 Prediction Error on Costs and Dynamical Matrices

We now consider prediction errors on cost functions and dynamics, rather than disturbances. Specifically, we consider the following instance of problem (1):

$$\min_{x_{0:T}, u_{0:T-1}} \sum_{t=0}^{T-1} \left((x_t - \bar{x}_t(\xi_t^*))^\top Q_t(\xi_t^*)(x_t - \bar{x}_t(\xi_t^*)) + u_t^\top R_t(\xi_t^*) u_t\right) + F_T(x_T; \xi_t^*)$$
$$\text{s.t. } x_{t+1} = A_t(\xi_t^*) \cdot x_t + B_t(\xi_t^*) \cdot u_t + w_t(\xi_t^*), \qquad \forall 0 \le t < T, \qquad (10)$$
$$x_0 = x(0),$$

where the terminal cost is given by $F_T(x_T; \xi_T^*) := (x_T - \bar{x}_T(\xi_T^*))^\top P_T(\xi_T^*)(x_T - \bar{x}(\xi_T^*))$.

All necessary assumptions on the system are summarized below in Assumption 4.2.

**Assumption 4.2.** *Assume the following holds for the online control problem instance* (10)*:*

- *Cost:* $\mu I \preceq Q_t(\xi_t) \preceq \ell I, \mu I \preceq R_t(\xi_t) \preceq \ell I,$ *and* $\mu I \preceq P_T(\xi_T) \preceq \ell I, \forall \xi_t \in \Xi_t, \forall t.$
- *Dynamical systems: both the ground-truth LTV system* $\{A_t(\xi_t^*), B_t(\xi_t^*)\}_{t=0}^{T-1}$ *and any predicted LTV system* $\{A_t(\xi_{t+\tau|t}), B_t(\xi_{t+\tau|t})\}_{\tau=0}^{k-1}$ *(for all* $\xi_t \in \Xi_t$ *and all* $t$*) satisfy the controllability assumptions in Assumption G.1 in Appendix G.*
- *Predicted quantities: bounds* $\|w_t(\xi_t)\| \leq D_w, \|\bar{x}_t(\xi_t)\| \leq D_{\bar{x}}, \|A_t(\xi_t)\| \leq a, \|B_t(\xi_t)\| \leq b$ *hold for all* $\xi_t \in \Xi_t$ *and all* $t$*.* $L_A$ *is a uniform Lipschitz constant such that* $\|A_t(\xi_t) - A_t(\xi_t')\| \leq L_A \|\xi_t - \xi_t'\|, \forall \xi_t, \xi_t' \in \Xi_t$ *holds for all* $t$*, and* $L_B, L_Q, L_R, L_{\bar{x}}, L_w$ *are defined similarly.*

Under Assumption 4.2, we can show that for some constants $H_2 \geq 1$ and $\lambda_2 \in (0, 1)$, perturbation bounds (3) and (4) hold globally for $q_1(t) = H_2\lambda_2^{2t}$, $q_2(t) = H_2\lambda_2^t$, and $q_3(t) = H_2\lambda_2^t$ under the specifications of Property 3.1. Thus, Property 3.1 holds for arbitrary $R$, and we can set $R = D_x^* + D_{\bar{x}}$ so that Theorem 3.3 can be applied to $\mathsf{MPC}_k$ with terminal cost $F_{t+k}(\cdot; \xi_{t|t+k}) = \mathbb{I}(\cdot; \bar{x}(\xi_{t|t+k}))$, which leads to the following dynamic regret bound:

**Theorem 4.2.** *In the unconstrained LTV setting* (10)*, under Assumption 4.2, when the prediction horizon* $k \geq O(1)$ [1] *and the prediction errors satisfy* $\sum_{\tau=0}^{k} \lambda_2^{2\tau} \rho_{t,\tau} \leq \Omega(1)$ *for all* $t$*, the dynamic regret of* $\mathsf{MPC}_k$ *(Algorithm 1) with terminal cost* $F_{t+k}(\cdot; \xi_{t|t+k}) = \mathbb{I}(\cdot; \bar{x}(\xi_{t|t+k}))$ *is bounded by*

$$\mathrm{cost}(\mathsf{MPC}_k) - \mathrm{cost}(\mathsf{OPT}) \leq O\left(\sqrt{T \cdot \sum_{\tau=0}^{k-1} \lambda_2^\tau P(\tau) + \lambda_2^{2k} T^2} + \sum_{\tau=0}^{k-1} \lambda_2^\tau P(\tau)\right).$$

The exact constants and a complete proof of Theorem 4.2 can be found in Appendix G. Compared with Theorem 4.1, Theorem 4.2 additionally requires the discounted total prediction errors $\sum_{\tau=0}^{k} \lambda_2^{2\tau} \rho_{t,\tau}$ to be less than or equal to some constant. This is actually expected, and emphasizes the critical difference between the prediction errors on dynamical matrices $(A_t, B_t)$ and the prediction errors on $w_t$, since an online controller cannot even stabilize the system when the predictions on $(A_t, B_t)$ can be arbitrarily bad. It is worth noting that Assumption 4.2 requires the uniform controllability to hold for the unknown ground-truth LTV dynamics and any predicted dynamics. The goal is to ensure the perturbation bounds for KKT matrix inverse hold in Lemma G.2. Intuitively, this assumption is necessary because otherwise the solution of MPC (by solving FTOCP induced by the predicted dynamics) can be unbounded. We provided two examples (Example G.4 and G.5) that satisfy Assumption 4.2 while the true dynamics are unknown.

## 5   General Dynamical Systems

We now move beyond unconstrained linear systems to constrained nonlinear systems given by the general online control problem (1) in Section 2. All necessary assumptions are summarized in Assumption H.1 in Appendix H. Perhaps surprisingly, decaying perturbation bounds can hold even in this case. In particular, using Theorem 4.5 in [50], we can show that there exists a small constant $R$ such that, for some constants $H_3 \geq 1$ and $\lambda_3 \in (0, 1)$, perturbation bounds (3) and (4) hold for $q_1(t) = 0$, $q_2(t) = H_3\lambda_3^t$, and $q_3(t) = H_3\lambda_3^t$. Thus, Property 3.1 holds (see Appendix H for formal statements) and we can apply Theorem 3.3 to obtain the following dynamic regret bound:

**Theorem 5.1.** *In the general system* (1)*, under Assumption H.1 in Appendix H, Property 3.1 holds for some positive constant* $R$ *and* $q_1(t) = 0$*,* $q_2(t) = H_3\lambda_3^t$*, and* $q_3(t) = H_3\lambda_3^t$*. Suppose the terminal cost* $F_{t+k}$ *of* $\mathsf{MPC}_k$ *is set to be the indicator function of some state* $\bar{y}(\xi_{t+k|t})$ *that satisfies* $\bar{y}(\xi_{t+k|t}) \in \mathcal{B}(x_{t+k}^*, R)$ *for* $t < T - k$*. Suppose the prediction errors* $\rho_{t,\tau}$ *are sufficiently small and the prediction horizon* $k$ *is sufficiently large such that* $H_3 \sum_{\tau=0}^{k-1} \lambda_3^\tau \rho_{t,\tau} + 2RH_3\lambda_3^k \leq \frac{(1-\lambda_3)^2 R}{H_3^2 L_g}$*. Then, the dynamic regret of* $\mathsf{MPC}_k$ *is upper bounded by* $\mathrm{cost}(\mathsf{MPC}_k) - \mathrm{cost}(\mathsf{OPT}) \leq$

$$O\left(\sqrt{T \cdot \sum_{\tau=0}^{k-1} \lambda_3^\tau P(\tau) + \lambda_3^{2k} T^2} + \sum_{\tau=0}^{k-1} \lambda_3^\tau P(\tau)\right).$$

A complete proof of Theorem 5.1 can be found in Appendix H. An assumption in Theorem 5.1 that is difficult to satisfy in general is that the reference terminal states $\bar{y}(\xi_{t+k|t})$ of $\mathsf{MPC}_k$ must be close enough to the offline optimal state $x_{t+k}^*$, i.e., $\bar{y}(\xi_{t+k|t}) \in \mathcal{B}(x_{t+k}^*, R)$, while the offline optimal state $x_{t+k}^*$ is generally unknown. This can be achieved in some special cases, for example, when we

---

[1]When we say $z \geq O(1)$, we mean there exists $c = O(1)$ such that $z \geq c$ holds.

know $\|\xi_t^*\|$ is sufficiently small. In this case, one can first solve FTOCP $\psi_0^T(x_0, \mathbf{0}; F_T)$ and use it as a reference to set the terminal states of $\mathsf{MPC}_k$. This intuition is formally shown in Appendix H. Another limitation is that Theorem 5.1 is only a bound on the cost of MPC, not its feasibility. There are many ways to guarantee recursive feasibility of MPC [53], which we leave as future work. We also discuss how to verify Assumption H.1 in two simple examples that arise from a simple inventory dynamics in Appendix I. The first positive example shows that Assumption H.1 is not vacuous, and the second negative example shows exponentially decaying perturbation bounds may not hold when Assumption H.1 is not satisfied.

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
