# A  Notation Summary

In this paper, we use $\alpha_{t_1:t_2}$ ($t_2 \geq t_1$) to denote a sequence of vectors $(\alpha_{t_1}, \alpha_{t_1+1}, \ldots, \alpha_{t_2})$. For ease of reference, we summarize in the following table all the notations used in the paper.

| Notation | Meaning |
|---|---|
| $\xi_t$ | The uncertainty parameter of the system, used to parameterize costs, dynamics, and constraints. |
| $\xi_t^*$ | The ground-truth parameter of the system, unknown to the controller. |
| $\xi_{\tau\|t}$ | The prediction of $\xi_\tau^*$ revealed to the controller at time step $t$ ($\tau \geq t$). |
| $\Xi_t$ | The space of uncertainty parameters. $\xi_t^*$ and $\xi_{t\|\tau}, \tau \leq t$ are in $\Xi_t$. We assume the diameter of $\Xi_t$ is less than or equal to 1 without the loss of generality, i.e., $\|\xi_t - \xi_t'\| \leq 1$ for all $\xi_t, \xi_t' \in \Xi_t$. |
| $k$ | The prediction horizon. At time $t$, the controller observes predictions $\xi_{t:t'\|t}$, where $t' := \min\{t+k, T\}$. |
| $\rho_{t,\tau}$ | The error of predicting the system parameter after $\tau$ steps at time $t$, i.e., $\rho_{t,\tau} = \left\|\xi_{t+\tau}^* - \xi_{t+\tau\|t}\right\|$. We adopt the convention that $\rho_{t,\tau} := 0$ if $t + \tau > T$. |
| $P(\tau)$ | The total error of predicting the system parameter after $\tau$ steps (the power of $\tau$-step-away predictions), i.e., $P(\tau) := \sum_{t=0}^{T-\tau} \rho_{t,\tau}^2$. |
| $f_t(x_t, u_t; \xi_t)$ | The stage cost of FTOCP at time step $t$, parameterized by $\xi_t \in \Xi_t$. The true stage cost is $f_t(x_t, u_t; \xi_t^*)$. |
| $g_t(x_t, u_t; \xi_t)$ | The dynamical function at time step $t$, parameterized by $\xi_t \in \Xi_t$. The true dynamics is $x_{t+1} = g_t(x_t, u_t; \xi_t^*)$. |
| $s_t(x_t, u_t; \xi_t)$ | The constraint function at time step $t$, parameterized by $\xi_t \in \Xi_t$. The true constraint is $s_t(x_t, u_t; \xi_t^*) \leq 0$. |
| $F_T$ and $\{F_{t+k}\}_{t=0}^{T-k-1}$ | $F_T$ is the true terminal cost function defined by the original online control problem (1), while $F_{t+k}$ for $t < T - k$ is the terminal cost function used by $\mathsf{MPC}_k$ at time $t$. |
| $\iota_{t_1}^{t_2}(z, \xi_{t_1:t_2-1}, \zeta_{t_2}; F)$ | The FTOCP defined on the time interval $[t_1, t_2]$, where $z$ is the initial state at time $t_1$, and $F$ is some terminal cost function at time $t_2$. $\xi_{t_1:t_2-1}$ are the parameters for the cost, dynamics, and constraints at time $[t_1, t_2 - 1]$, while $\zeta_{t_2}$ is the parameter for the terminal cost $F$. |
| $\psi_{t_1}^{t_2}(z, \xi_{t_1:t_2-1}, \zeta_{t_2}; F)$ | An optimal solution to the FTOCP $\iota_{t_1}^{t_2}(z, \xi_{t_1:t_2-1}, \zeta_{t_2}; F)$. The entries are indexed by $y_{t_1:t_2}$ (for states) and $v_{t_1:t_2-1}$ (for actions). |
| $\psi_{t_1}^{t_2}(z, \xi_{t_1:t_2}; F)$ | The shorthand notation of $\psi_{t_1}^{t_2}(z, \xi_{t_1:t_2-1}, \xi_{t_2}; F)$. |

# B  Assumptions Overview

In this section, we give a more detailed overview of the assumptions that the online control problem (1) should satisfy in general so that our pipeline in Section 3.2 works. Specific assumptions in each specific setting will be presented separately in Assumption F.1, G.1, and H.1.

**Cost functions.** In general, we require the stage cost functions $f_t$ and the terminal cost $F_T$ to be *well-conditioned*, which includes non-negativity, strong convexity, smoothness (Lipschitz continuous gradient), and twice continuous differentiability. Note that these assumptions are equivalent to bounded Hessian ($\mu I \preceq \nabla^2 f_t \preceq \ell I$) and non-negative minimizer of the cost functions. Specifically, for quadratic costs $\nabla^2 f_t$ are constant, and the assumptions are further equivalent to bounded spectra of the cost matrices.

**Dynamical systems.** A basic requirement of the dynamical function $g_t$ is *Lipschitzness* in $u_t$, i.e.,

$$\|g_t(x_t, u_t; \xi_t^*) - g_t(x_t, u_t'; \xi_t^*)\| \leq L_g \|u_t - u_t'\|.$$

We point out that only Lipschitzness in control action $u_t$ is needed for the Pipeline Theorem to hold, which guarantees that an error on a control action $u_t$ has a bounded impact on the next state $x_{t+1}$.

A more non-trivial assumption on dynamics is that the dynamical system should be *(uniformly) controllable*. Intuitively, this means the online controller should be able to steer the system to some target state in a finite number of time steps with some bounded control actions.

**Definition B.1** (uniform controllability). *Consider a general dynamics $x_{t+1} = g_t(x_t, u_t; \xi_t)$. For any time steps $t_2 \geq t_1$ and fixed $(x_t, u_t)$, define $A_t := \nabla_{x_t}^\top g_t(x_t, u_t; \xi_t)$ and $B_t := \nabla_{u_t}^\top g_t(x_t, u_t; \xi_t)$, and we further define **transition matrix** $\Phi(t_2, t_1) \in \mathbb{R}^{n \times n}$ at $(x_t, u_t)$ as*

$$\Phi(t_2, t_1) := \begin{cases} A_{t_2-1} A_{t_2-2} \cdots A_{t_1} & \text{if } t_2 > t_1, \\ I & \text{otherwise.} \end{cases}$$

*For any time $t$ and time interval $p \geq 0$, define **controllability matrix** $M(t, p; x_{t:t+p}, u_{t:t+p}) \in \mathbb{R}^{n \times (mp)}$ as*

$$M(t, p; x_{t:t+p}, u_{t:t+p}) := [\Phi(t+p, t+1)B_t, \Phi(t+p, t+2)B_{t+1}, \ldots, \Phi(t+p, t+p)B_{t+p}].$$

*We say the system is **controllable** if there exists a positive integer $d$, such that the controllability matrix $M(t, d; x_t, u_t)$ is of full row rank for any $t$ and any $(x_t, u_t)$. The smallest such constant $d$ is called the **controllability index** of the system. Further, we say the system is **$\sigma$-uniformly controllable** if exists a positive constant $\sigma$ such that $\sigma_{\min}(M(t, d)) \geq \sigma$ holds for all $t = 0, \ldots, T - d$.*

The definition has a clear control-theoretic interpretation for linear dynamics (where $A_t$ and $B_t$ are independent of $(x_t, u_t)$), but might seem trickier for non-linear dynamics (where $A_t$ and $B_t$ are functions of $(x_t, u_t)$). For the latter case, uniform controllability may be assumed for the offline optimal trajectory only, or for state-action pairs in a small neighborhood around it.

**Constraints.** Recursive feasibility is a well-known challenge for the design of online controllers in constrained systems [53]: at some time $t$, the controller may encounter an absence of feasible trajectories to continue from the current state $x_t$. Many solutions have been proposed for different controllers in a variety of systems. Since the purpose of this work is to establish dynamic regret guarantees for an online controller, and for the purpose of this paper, we would expect that there is a solution, potentially via a combination of proper controller design (e.g., setting the terminal cost/constraint of MPC) and some additional assumptions on the system (e.g., the SSOSC, strong second-order sufficient conditions, and LICQ, linear independent constraint qualification, which will be introduced in Section 5), so that we could focus on the sub-optimality of the online controller against the offline optimal trajectory.

We also need to point out that, although the additional assumptions on system that involve constraints might seem tricky, sometimes they are exactly the implications of previous assumptions on costs and dynamics that is actually needed in the proof. For example, Lemma 12 in [37] shows that Lipshitzness of dynamics and uniform controllability together imply uniform LICQ property of the system. For the clarity of exposition, these implications might be directly assumed in place of the low-level ones.

**Parameter $\xi_t$.** In general, we require that all predicted quantities, which might include cost functions, dynamical functions, and constraints, should be *Lipschitz* in $\xi_t$, so that these quantities get closer to their ground truth value in a linearly-bounded way as the prediction error on the parameter $\xi_t^*$ decreases. For a specific example of parameterized linear dynamics $x_{t+1} = A_t(\xi_t)x_t + B_t(\xi_t)u_t + w_t(\xi_t)$, the requirement is realized by assuming Lipschitzness of $A_t(\cdot), B_t(\cdot), w_t(\cdot)$ in $\xi_t$.

**Offline optimal trajectory.** We require the offline optimal trajectory OPT to be *stable*; i.e., there exists a constant $D_{x^*}$ such that $\|x_t^*\| \leq D_{x^*}$ for any state $x_t^*$ visited by OPT. While this can be shown under some assumptions in unconstrained LTV systems (see [35]), we introduce this assumption to simplify and unify the presentation for more complex systems.

## C  Proof of Lemma 3.1

We have already shown (5) holds for all time step $t < T - k$ in the main body. For $t \geq T - k$, we see that

$$e_t = \left\| \psi_t^T \left( x_t, \xi_{t:T|t}; F_T \right) - \psi_t^T \left( x_t, \xi_{t:T}^*; F_T \right) \right\| \tag{11a}$$

$$\leq \sum_{\tau=0}^{k} \left( \|x_t\| \cdot q_1(\tau) + q_2(\tau) \right) \rho_{t,\tau} \tag{11b}$$

$$\leq \sum_{\tau=0}^{k} \left( \left( \frac{R}{C_3} + D_{x^*} \right) \cdot q_1(\tau) + q_2(\tau) \right) \rho_{t,\tau}, \tag{11c}$$

where we used the definition of per-step error $e_t$ in (11a); we used the perturbation bound (3) specified by Property 3.1 in (11b); we used the assumption $x_t \in \mathcal{B}\left(x_t^*, \frac{R}{C_3}\right)$, $\|x_t^*\| \le D_{x^*}$, and the convention $\rho_{t,\tau} := 0$ if $t + \tau > T$ in (11c). Thus $e_t$ also satisfies (5) for $t \ge T - k$.

## D    Proof of Lemma 3.2

To simplify the notation, we will use $\psi_t^T(z)$ as a shorthand notation of $\psi_t^T(z, \xi_{t:T}^*; F_T)$ in the proof of Lemma 3.2, since the proof only relies on the perturbation bound (4).

Note that for any time step $t + 1$, by Lipschitzness of the dynamics we have

$$
\begin{aligned}
\left\| x_{t+1} - \psi_t^T(x_t)_{y_{t+1}} \right\| &= \left\| g_t(x_t, u_t, w_t) - g_t\left(x_t, \psi_t^T(x_t)_{v_t}, w_t\right) \right\| \\
&\le L_g \left\| u_t - \psi_t^T(x_t)_{v_t} \right\| \\
&\le L_g e_t.
\end{aligned}
\tag{12}
$$

Therefore, we can show the statement that $x_t \in \mathcal{B}\left(x_t^*, \frac{R}{C_3}\right)$ holds if $e_\tau \le R/(C_3^2 L_g), \forall \tau < t$ by induction. Note that this statement clearly holds for $t = 0$ since $x_0^* = x_0$. Suppose it holds for $0, 1, \ldots, t - 1$. Then, we see that

$$
\begin{aligned}
\|x_t - x_t^*\| &= \left\| x_t - \psi_0^T(x_0)_{y_t} \right\| \\
&\le \left\| x_t - \psi_{t-1}^T(x_{t-1})_{y_t} \right\| + \sum_{i=1}^{t-1} \left\| \psi_{t-i}^T(x_{t-i})_{y_t} - \psi_{t-i-1}^T(x_{t-i-1})_{y_t} \right\| \\
&\le \left\| x_t - \psi_{t-1}^T(x_{t-1})_{y_t} \right\| + \sum_{i=1}^{t-1} q_3(i) \left\| x_{t-i} - \psi_{t-i-1}^T(x_{t-i-1})_{y_{t-i}} \right\| \\
&\le \sum_{i=0}^{t-1} q_3(i) \left\| x_{t-i} - \psi_{t-i-1}^T(x_{t-i-1})_{y_{t-i}} \right\| \\
&\le L_g \sum_{i=0}^{t-1} q_3(i) e_{t-i-1},
\end{aligned}
$$

$$\tag{13a}$$
$$\tag{13b}$$
$$\tag{13c}$$

where in (13a), we apply the perturbation bound (4) specified by Property 3.1. To see why it can be applied, note that for $i \in [1, t-1]$, $x_{t-i-1}$ satisfies $x_{t-i-1} \in \mathcal{B}\left(x_{t-i-1}^*, \frac{R}{C_3}\right)$ by the induction assumption, thus we have $\psi_{t-i-1}^T(x_{t-i-1})_{y_{t-i}} \in \mathcal{B}\left(x_{t-i}^*, R\right)$ because $q_3(1) \le \sum_{\tau=0}^{\infty} q_3(\tau) \le C_3$. Therefore, we can apply the perturbation bound (4) specified by Property 3.1 to compare the optimization solution vectors $\psi_{t-i}^T(x_{t-i})$ and $\psi_{t-i}^T\left(\psi_{t-i-1}^T(x_{t-i-1})_{y_{t-i}}\right)$, and by the principle of optimality, we see that

$$
\psi_{t-i}^T\left(\psi_{t-i-1}^T(x_{t-i-1})_{y_{t-i}}\right)_{y_t} = \psi_{t-i-1}^T(x_{t-i-1})_{y_t}.
$$

We also used $q_3(0) \ge 1$ in (13b) and (12) in (13c). Recall that we assume $e_{t-i} \le \frac{R}{C_3^2 L_g}$. Substituting this into (13) gives that

$$
\|x_t - x_t^*\| \le L_g \cdot \frac{R}{C_3^2 L_g} \sum_{i=0}^{t-1} q_3(i) \le \frac{R}{C_3}.
$$

Hence we have shown $x_t \in \mathcal{B}\left(x_t^*, \frac{R}{C_3}\right)$ holds if $e_\tau \le R/(C_3^2 L_g), \forall \tau < t$ by induction. An implication of this result is that $x_t \in \mathcal{B}\left(x_t^*, \frac{R}{C_3}\right)$ holds for all $t \le T$ if $e_t \le R/(C_3^2 L_g)$ holds for all $t < T$.

Similar with (13), we see the following inequality holds for all $t < T$ if $e_t \le R/(C_3^2 L_g), \forall t < T$:

$$
\|u_t - u_t^*\| = \left\| u_t - \psi_0^T(x_0)_{v_t} \right\|
$$

$$\leq \left\| u_t - \psi_t^T(x_t)_{v_t} \right\| + \sum_{i=0}^{t-1} \left\| \psi_{t-i}^T(x_{t-i})_{v_t} - \psi_{t-i-1}^T(x_{t-i-1})_{v_t} \right\|$$

$$\leq \left\| u_t - \psi_t^T(x_t)_{v_t} \right\| + \sum_{i=0}^{t-1} q_3(i) \left\| x_{t-i} - \psi_{t-i-1}^T(x_{t-i-1})_{y_{t-i}} \right\|$$

$$\leq e_t + L_g \sum_{i=0}^{t-1} q_3(i) e_{t-i-1}, \tag{14}$$

where the second inequality holds for the same reason as (13a).

By (13), we see that

$$\|x_t - x_t^*\|^2 \leq L_g^2 \left( \sum_{i=0}^{t-1} q_3(i) e_{t-i-1} \right)^2$$

$$\leq L_g^2 \left( \sum_{i=0}^{t-1} q_3(i) \right) \cdot \left( \sum_{i=0}^{t-1} q_3(i) e_{t-i-1}^2 \right) \tag{15a}$$

$$\leq C_3 L_g^2 \left( \sum_{i=0}^{t-1} q_3(i) e_{t-i-1}^2 \right), \tag{15b}$$

where we use the Cauchy-Schwarz inequality in (15a), and $\sum_{i=0}^{t-1} q_3(i) \leq C_3$ in (15b).

Similarly, by (14), we see that

$$\|u_t - u_t^*\|^2 \leq \left( e_t + L_g \sum_{i=0}^{t-1} q_3(i) e_{t-i-1} \right)^2$$

$$\leq \left( 1 + L_g^2 \sum_{i=0}^{t-1} q_3(i) \right) \cdot \left( e_t^2 + \sum_{i=0}^{t-1} q_3(i) e_{t-i-1}^2 \right) \tag{16a}$$

$$\leq \left( 1 + C_3 L_g^2 \right) \cdot \left( e_t^2 + \sum_{i=0}^{t-1} q_3(i) e_{t-i-1}^2 \right), \tag{16b}$$

where we use the Cauchy-Schwarz inequality in (16a), and we use $\sum_{i=0}^{t-1} q_3(i) \leq C_3$ in (16b).

Summing (15) and (16) over time steps $t$ gives that

$$\sum_{t=1}^{T} \|x_t - x_t^*\|^2 + \sum_{t=0}^{T-1} \|u_t - u_t^*\|^2$$

$$\leq C_3 L_g^2 \sum_{t=1}^{T} \left( \sum_{i=0}^{t-1} q_3(i) e_{t-i-1}^2 \right) + \left( 1 + C_3 L_g^2 \right) \cdot \sum_{t=0}^{T-1} \left( e_t^2 + \sum_{i=0}^{t-1} q_3(i) e_{t-i-1}^2 \right)$$

$$\leq \left( 1 + 2 C_3 L_g^2 \right) \cdot \left( 1 + C_3 \right) \cdot \sum_{t=0}^{T-1} e_t^2, \tag{17}$$

where we rearrange the terms and use $\sum_{j=0}^{\infty} q_3(j) \leq C_3$ in the last inequality.

Since the cost function $f_t(\cdot, \cdot; \xi_t^*)$ and $F_T(\cdot; \xi_T^*)$ are nonnegative, convex, and $\ell$-smooth in their inputs, by Lemma F.2 in [35], we see that the following inequality holds for arbitrary $\eta > 0$:

$$\text{cost}(\mathsf{ALG}) - \text{cost}(\mathsf{OPT})$$

$$\leq \left( \sum_{t=0}^{T-1} f_t(x_t, u_t; \xi_t^*) + F_T(x_T; \xi_T^*) \right) - \left( \sum_{t=0}^{T-1} f_t(x_t^*, u_t^*; \xi_t^*) + F_T(x_T^*; \xi_T^*) \right)$$

$$\leq \eta \left( \sum_{t=0}^{T-1} f_t(x_t^*, u_t^*; \xi_t^*) + F_T(x_T^*; \xi_T^*) \right)$$

$$+ \frac{\ell}{2} \left(1 + \frac{1}{\eta}\right) \left(\sum_{t=1}^{T} \|x_t - x_t^*\|^2 + \sum_{t=0}^{T-1} \|u_t - u_t^*\|^2\right) \tag{18a}$$

$$\leq \eta \cdot \text{cost}(\text{OPT}) + \left(1 + \frac{1}{\eta}\right) \cdot \frac{\ell}{2} \cdot \left(1 + 2C_3 L_g^2\right) \cdot (1 + C_3) \cdot \sum_{t=0}^{T-1} e_t^2 \tag{18b}$$

$$= \eta \cdot \text{cost}(\text{OPT}) + \frac{1}{\eta} \cdot \frac{\ell}{2} \cdot \left(1 + 2C_3 L_g^2\right) \cdot (1 + C_3) \cdot \sum_{t=0}^{T-1} e_t^2$$

$$+ \frac{\ell}{2} \cdot \left(1 + 2C_3 L_g^2\right) \cdot (1 + C_3) \cdot \sum_{t=0}^{T-1} e_t^2, \tag{18c}$$

where we apply Lemma F.2 in [35] in (18a), and we use (17) in (18b). Setting the tunable weight $\eta$ in (18c) to be

$$\eta = \left(\frac{\frac{\ell}{2} \cdot \left(1 + 2C_3 L_g^2\right) \cdot (1 + C_3) \cdot \sum_{t=0}^{T-1} e_t^2}{\text{cost}(\text{OPT})}\right)^{\frac{1}{2}}$$

gives that

$$\text{cost}(\text{ALG}) - \text{cost}(\text{OPT})$$

$$\leq \sqrt{\left(\frac{\ell}{2} \cdot \left(1 + 2C_3 L_g^2\right) \cdot (1 + C_3)\right) \cdot \text{cost}(\text{OPT}) \cdot \sum_{t=0}^{T-1} e_t^2}$$

$$+ \frac{\ell}{2} \cdot \left(1 + 2C_3 L_g^2\right) \cdot (1 + C_3) \cdot \sum_{t=0}^{T-1} e_t^2. \tag{19}$$

This finishes the proof of Lemma 3.2.

## E   Proof of Theorem 3.3

We first use induction to show that the following two conditions holds for all time steps $t < T$:

$$x_t \in \mathcal{B}\left(x_t^*, \frac{R}{C_3}\right), \tag{20a}$$

$$e_t \leq \sum_{\tau=0}^{k} \left(\left(\frac{R}{C_3} + D_{x^*}\right) \cdot q_1(\tau) + q_2(\tau)\right) \rho_{t,\tau} + 2R\left(\left(\frac{R}{C_3} + D_{x^*}\right) \cdot q_1(k) + q_2(k)\right). \tag{20b}$$

At time step 0, (20a) holds because $x_0 = x_0^*$, and (20b) holds by Lemma 3.1 and the assumption on the terminal cost $F_k$ of $\text{MPC}_k$.

Suppose (20a) and (20b) hold for all time steps $\tau < t$. For time step $t$, by the assumption on the prediction errors $\rho_{t,\tau}$ and prediction horizon $k$ in Theorem 3.3, we know that $e_\tau \leq \frac{R}{C_3^2 L_g}$ holds for all $\tau < t$ because (20b) holds for all $\tau < t$. Thus, we know that (20a) holds for time step $t$ by Lemma 3.2. Then, since (20a) holds for time step $t$, and the terminal cost $F_{t+k}$ of $\text{MPC}_k$ is set to be the indicator function of some state $\bar{y}(\xi_{t+k|t})$ that satisfies $\bar{y}(\xi_{t+k|t}) \in \mathcal{B}(x_{t+k}^*, R)$ if $t < T - k$, we know (20b) also holds for time step $t$ by Lemma 3.1. This finishes the induction proof of (20).

To simplify the notation, let $R_0 := \frac{R}{C_3} + D_{x^*}$. Note that (20b) implies that

$$e_t^2 \leq \left(\sum_{\tau=0}^{k} (R_0 \cdot q_1(\tau) + q_2(\tau)) + 2R(R_0 + 1)\right)$$

$$\cdot \left(\sum_{\tau=0}^{k} (R_0 \cdot q_1(\tau) + q_2(\tau)) \rho_{t,\tau}^2 + 2R\left(R_0 \cdot q_1(k)^2 + q_2(k)^2\right)\right) \tag{21a}$$

$$\leq (R_0 C_1 + C_2 + 2R(R_0 + 1))$$

$$\cdot \left( \sum_{\tau=0}^{k-1} \left( R_0 \cdot q_1(\tau) + q_2(\tau) \right) \rho_{t,\tau}^2 + (2R+1) \left( R_0 \cdot q_1(k)^2 + q_2(k)^2 \right) \right), \tag{21b}$$

where we use the Cauchy-Schwarz inequality in (21a); we use the bounds $\sum_{\tau=0}^{k} q_1(\tau) \leq C_1$, $\sum_{\tau=0}^{k} q_2(\tau) \leq C_2$, and $\rho_{t,\tau} \leq 1$ in (21b).

Since (20) and (21) holds for all time steps $t < T$, we can apply Lemma 3.2 to obtain that

$$\text{cost}(\mathsf{MPC}_k) - \text{cost}(\mathsf{OPT}) \leq \sqrt{\text{cost}(\mathsf{OPT}) \cdot E_0} + E_0,$$

where

$$E_0 := (R_0 C_1 + C_2 + 2R(R_0 + 1))$$
$$\cdot \left( \sum_{\tau=0}^{k-1} \left( R_0 \cdot q_1(\tau) + q_2(\tau) \right) P(\tau) + (2R+1) \left( R_0 \cdot q_1(k)^2 + q_2(k)^2 \right) T \right).$$

This finishes the proof of Theorem 3.3.

## F  Assumptions and Proofs of Section 4.1

The formal definition of the controllability index $d$ and $\sigma$-uniform controllable are given in [35]. For completeness, we restate them for LTV dynamics in Assumption F.1 below.

**Assumption F.1.** *For time steps $t_2 \geq t_1$, we define the transition matrix $\Phi(t_2, t_1) \in \mathbb{R}^{n \times n}$ as*

$$\Phi(t_2, t_1) := \begin{cases} A_{t_2-1} A_{t_2-2} \cdots A_{t_1} & \text{if } t_2 > t_1 \\ I & \text{otherwise.} \end{cases},$$

*For any positive integer $p$, we define the controllability matrix $M(t, p) \in \mathbb{R}^{n \times (mp)}$ as*

$$M(t, p) := \left[ \Phi(t+p, t+1) B_t, \Phi(t+p, t+2) B_{t+1}, \dots, \Phi(t+p, t+p) B_{t+p} \right].$$

*We assume the LTV system $\{A_t, B_t\}$ is $\sigma$-uniform controllable with controllability index $d$, i.e., $d$ is the smallest positive integer such that $\sigma_{min}(M(t, d)) > 0$ holds for all $t \in [0, T-d]$, and $\sigma_{min}(M(t, d)) \geq \sigma$ holds for all $t \in [0, T-d]$.*

As a remark, the Assumption F.1 is a special case of Definition B.1 in unconstrained LTV systems. [35] has established a perturbation bound for the LTV system in (9) which implies the our requirements in Property 3.1. Thus we can use Theorem 3.3 to show Theorem 4.1.

*Proof of Theorem 4.1.* By Theorem 3.3 in [35], we know Property 3.1 holds under Assumption 4.1 for arbitrary $R$ and $q_1(t) = 0, q_2(t) = H_1 \lambda_1^t$, and $q_3(t) = H_1 \lambda_1^t$, where $H_1 = H_1(\mu, \ell, d, \sigma, a, b, b', L_w) > 0$ is some constant, and $\lambda_1 = \lambda_1(\mu, \ell, d, \sigma, a, b, b', L_w) \in (0, 1)$ is the decay rate. Here, $H_1$ corresponds to $C$ and $\lambda_1$ corresponds to $\lambda$ in Theorem 3.3 in [35].

By setting $R := \max\left\{ D_{x^*}, \frac{2L_g H_1^3}{(1-\lambda_1)^3} \right\}$, we guarantee that the terminal state $0$ of $\mathsf{MPC}_k$ is always in the closed ball $\mathcal{B}(x_{t+k}^*, R)$, and the condition

$$\sum_{\tau=0}^{k} \left( \left( \frac{R}{C_3} + D_{x^*} \right) \cdot q_1(\tau) + q_2(\tau) \right) \rho_{t,\tau} + 2R \left( \left( \frac{R}{C_3} + D_{x^*} \right) \cdot q_1(k) + q_2(k) \right) \leq \frac{R}{C_3^2 L_g}$$

holds once $k \geq \ln\left( \frac{4H_1^3 L_g}{(1-\lambda_1)^2} \right) / \ln(1/\lambda_1)$ because $\rho_{t,\tau} \leq 1$. Therefore, we can apply Theorem 3.3 to finish the proof of Theorem 4.1. $\qquad\square$

## G  Assumptions and Proofs of Section 4.2

In this section, we give the detailed assumptions and proofs of the results in Section 4.2. Before we present the assumption on the uncertain LTV systems in (10), we first define several quantities that we will use heavily in the rest of this section:

For time steps $t_1 \leq t_2$ and $\xi_{t_1:t_2} \in \Xi_{t_1:t_2}$, define

$$N_{t_1}^{t_2}(\xi_{t_1:t_2}) := \begin{bmatrix} I \\ -A_{t_1}(\xi_{t_1}) & -B_{t_1}(\xi_{t_1}) & I \\ & & \ddots \\ & & & -A_{t_2}(\xi_{t_2}) & -B_{t_2}(\xi_{t_2}) & I \end{bmatrix}. \qquad (22)$$

This matrix is closely related to the stability of the LTV system in the time interval $[t_1, t_2 + 1]$. To see this, note that $N_{t_1}^{t_2}(\xi_{t_1:t_2})$ always has full row rank, i.e., given any disturbance vector $w = (x_{t_1}, w_{t_1}, w_{t_1+1}, \ldots, w_{t_2})^\top$, one can always find a feasible sub-trajectory $z = (x_{t_1}, u_{t_1}, x_{t_1+1}, \ldots, u_{t_2}, x_{t_2+1})^\top$ that satisfies $N_{t_1}^{t_2}(\xi_{t_1:t_2})z = w$. If for any vector $w$, there exists a feasible sub-trajectory $z$ such that $\|z\| \leq (1/\sigma) \cdot \|w\|$ for some positive constant $\sigma$, then the smallest singular value of $N_{t_1}^{t_2}(\xi_{t_1:t_2})$ is lower bounded by $\sigma$.

Similar with (22), we define matrix

$$\hat{N}_{t_1}^{t_2}(\xi_{t_1:t_2}) := \begin{bmatrix} I \\ -A_{t_1}(\xi_{t_1}) & -B_{t_1}(\xi_{t_1}) & I \\ & & \ddots \\ & & & -A_{t_2}(\xi_{t_2}) & -B_{t_2}(\xi_{t_2}) \end{bmatrix} \qquad (23)$$

for any time steps $t_1 \leq t_2$ and $\xi_{t_1:t_2} \in \Xi_{t_1:t_2}$, which removes the last column of (22). The matrix $\hat{N}_{t_1}^{t_2}(\xi_{t_1:t_2})$ is closely related to the controllability of the LTV system in the time interval $[t_1, t_2 + 1]$. To see this, given any disturbance vector $\hat{w} = (x_{t_1}, w_{t_1}, w_{t_1+1}, \ldots, w_{t_2} - x_{t_2+1})^\top$ whose first/last entry depends on the initial/terminal state, a feasible sub-trajectory $\hat{z} = (x_{t_1}, u_{t_1}, x_{t_1+1}, \ldots, u_{t_2})^\top$ must satisfy that $\hat{N}_{t_1}^{t_2}(\xi_{t_1:t_2})\hat{z} = \hat{w}$. Different from $N_{t_1}^{t_2}(\xi_{t_1:t_2})$, $\hat{N}_{t_1}^{t_2}(\xi_{t_1:t_2})$ is not guaranteed to have full row rank. If for any vector $\hat{w}$, there exists a feasible sub-trajectory $\hat{z}$ such that $\|\hat{z}\| \leq (1/\sigma) \cdot \|\hat{w}\|$ for some positive constant $\sigma$, then the smallest singular value of $\hat{N}_{t_1}^{t_2}(\xi_{t_1:t_2})$ is lower bounded by $\sigma$.

We make the following assumption on the smallest singular values of matrices $N_{t_1}^{t_2}(\xi_{t_1:t_2})$ and $\hat{N}_{t_1}^{t_2}(\xi_{t_1:t_2})$ so that the LTV system possesses uniform stability and controllability properties under any uncertainty parameters:

**Assumption G.1.** *There exists some universal constant $\sigma > 0$ such that $\sigma_{min}\left(N_t^{T-1}(\xi_{t:T-1})\right) \geq \sigma$ for any $t < T$, and $\sigma_{min}\left(\hat{N}_t^{t+k}\left(\xi_{t:t+k}\right)\right) \geq \sigma$ for any $t < T - k$.*

While Assumption G.1 may seem more restricted than the uniform controllability defined in Definition B.1, it can actually be derived from Definition B.1 by Lemma 12 in [37].

In order to formulate (10) as a quadratic programming problem with equality constraints, we also need to define the matrix for cost functions:

$$M_t^T(\xi_{t:T}) := \mathrm{diag}\left(Q_t(\xi_t), R_t(\xi_t), Q_{t+1}(\xi_{t+1}), \ldots, R_{T-1}(\xi_{T-1}), P_T(\xi_T)\right), \forall t < T,$$

$$\hat{M}_t^{t+k}(\xi_{t:t+k}) := \mathrm{diag}\left(Q_t(\xi_t), R_t(\xi_t), Q_{t+1}(\xi_{t+1}), \ldots, R_{t+k-1}(\xi_{t+k-1})\right), \forall t < T - k. \quad (24)$$

To write down the KKT condition of the equality constrained quadratic programming problem, we also need to define

$$H_t^T(\xi_{t:T}) := \begin{bmatrix} M_t^T(\xi_{t:T}) & N_t^{T-1}(\xi_{t:T-1})^\top \\ N_t^{T-1}(\xi_{t:T-1}) & 0 \end{bmatrix},$$

$$\hat{H}_t^{t+k}(\xi_{t:t+k}) := \begin{bmatrix} \hat{M}_t^{t+k}(\xi_{t:t+k}) & \hat{N}_t^{t+k-1}(\xi_{t:t+k-1})^\top \\ \hat{N}_t^{t+k-1}(\xi_{t:t+k-1}) & 0 \end{bmatrix},$$

$$b_t^T(z, \xi_{t:T}) := \left(Q_t(\xi_t)\bar{x}_t(\xi_t), 0, \ldots, P(\xi_T)\bar{x}_T(\xi_T), z, w_t(\xi_t), \ldots, w_{T-1}(\xi_{T-1})\right)^\top,$$

$$\hat{b}_t^{t+k}(z, \xi_{t:t+k}) := \left(Q_t(\xi_t)\bar{x}_t(\xi_t), 0, \ldots, 0, z, w_t(\xi_t), \ldots, w_{t+k-1}(\xi_{t+k-1}) - \xi_{t+k}\right)^\top,$$

$$\chi_t^T = \left(y_t, v_t, y_{t+1}, \ldots, v_{T-1}, y_T, \eta_t, \eta_{t+1}, \ldots, \eta_T\right)^\top,$$

$$\hat{\chi}_t^{t+k} = \left(y_t, v_t, y_{t+1}, \ldots, v_{t+k-1}, \eta_t, \eta_{t+1}, \ldots, \eta_{t+k}\right)^\top.$$

According to the KKT condition, the optimal primal-dual solution to $\iota_t^T(z, \xi_{t:T}; F_T)$ $(t < T)$ is the unique solution $\chi_t^T$ to the linear equation $H_t^T(\xi_{t:T})\chi_t^T = b_t^T(z, \xi_{t:T})$. Similarly, the optimal primal-dual solution to $\iota_t^{t+k}(z, \xi_{t:t+k}; \mathbb{I})$ $(t < T-k)$ is the unique solution $\chi_t^{t+k}$ to the linear equation $\hat H_t^{t+k}(\xi_{t:t+k})\hat\chi_t^{t+k} = \hat b_t^{t+k}(z, \xi_{t:t+k})$. We provide an illustrative example for $\chi_t^T$ with $(t,T) = (0,3)$ below:

$$
\left[\begin{array}{ccccccc|cccc}
Q_0 & & & & & & & I & -A_0^\top & & \\
& R_0 & & & & & & & -B_0^\top & & \\
& & Q_1 & & & & & & I & -A_1^\top & \\
& & & R_1 & & & & & & -B_1^\top & \\
& & & & Q_2 & & & & & I & -A_2^\top \\
& & & & & R_2 & & & & & -B_2^\top \\
& & & & & & P_3 & & & & I \\
\hline
I & & & & & & & & & & \\
-A_0 & -B_0 & I & & & & & & & & \\
& & -A_1 & -B_1 & I & & & & & & \\
& & & & -A_2 & -B_2 & I & & & &
\end{array}\right]
\left[\begin{array}{c} y_0\\v_0\\y_1\\v_1\\y_2\\v_2\\y_3\\\hline \eta_0\\\eta_1\\\eta_2\\\eta_3\end{array}\right]
=
\left[\begin{array}{c} Q_0\bar x_0\\0\\Q_1\bar x_1\\0\\Q_2\bar x_2\\0\\P_3\bar x_3\\\hline z\\w_0\\w_1\\w_2\end{array}\right],
$$

where we omit the parameters $\xi_{0:3}$ to simplify the notations. Rearranging the rows and columns of the matrix on the left hand side gives the equation:

$$
\left[\begin{array}{ccc|ccc|ccc|cc}
Q_0 & & I & & & -A_0^\top & & & & & \\
& R_0 & & & & -B_0^\top & & & & & \\
I & & & & & & & & & & \\
\hline
& & & Q_1 & & I & & & -A_1^\top & & \\
& & & & R_1 & & & & -B_1^\top & & \\
-A_0 & -B_0 & & I & & & & & & & \\
\hline
& & & & & & Q_2 & & I & & -A_2^\top \\
& & & & & & & R_2 & & & -B_2^\top \\
& & & -A_1 & -B_1 & & I & & & & \\
\hline
& & & & & & & & & P & I \\
& & & & & & -A_2 & -B_2 & & I &
\end{array}\right]
\left[\begin{array}{c} y_0\\v_0\\\eta_0\\y_1\\v_1\\\eta_1\\y_2\\v_2\\\eta_2\\y_3\\\eta_3\end{array}\right]
=
\left[\begin{array}{c} Q_0\bar x_0\\0\\z\\Q_1\bar x_1\\0\\w_0\\Q_2\bar x_2\\0\\w_1\\P\bar x_3\\w_2\end{array}\right].
$$

Let $\Phi_t^T$ denote the permutation matrix that permute $(y_t, v_t, y_{t+1}, \ldots, v_{T-1}, y_T; \eta_t, \ldots, \eta_T)^\top$ to $(y_t, v_t, \eta_t, y_{t+1}, v_{t+1}, \eta_{t+1}, \ldots, y_T, \eta_T)^\top$. We use $\Upsilon_t^T(\xi_{t:T}) := (\Phi_t^T)H_t^T(\xi_{t:T})(\Phi_t^T)^\top$ to denote the rearrangement of $H_t^T(\xi_{t:T})$ as illustrated in the above equation, and use $\beta_t^T(z, \xi_{t:T}) := (\Phi_t^T)b_t^T(z, \xi_{t:T})$ to denote the corresponding rearrangement of $b_t^T(z, \xi_{t:T})$.

We also provide an illustrative example for $\hat\chi_t^{t+k}$ with $(t, k) = (0, 3)$ below:

$$
\left[\begin{array}{cccccc|cccc}
Q_0 & & & & & & I & -A_0^\top & & \\
& R_0 & & & & & & -B_0^\top & & \\
& & Q_1 & & & & & I & -A_1^\top & \\
& & & R_1 & & & & & -B_1^\top & \\
& & & & Q_2 & & & & I & -A_2^\top \\
& & & & & R_2 & & & & -B_2^\top \\
\hline
I & & & & & & & & & \\
-A_0 & -B_0 & I & & & & & & & \\
& & -A_1 & -B_1 & I & & & & & \\
& & & & -A_2 & -B_2 & & & &
\end{array}\right]
\left[\begin{array}{c} y_0\\v_0\\y_1\\v_1\\y_2\\v_2\\\hline \eta_0\\\eta_1\\\eta_2\\\eta_3\end{array}\right]
=
\left[\begin{array}{c} Q_0\bar x_0\\0\\Q_1\bar x_1\\0\\Q_2\bar x_2\\0\\\hline z\\w_0\\w_1\\w_2\end{array}\right],
$$

where we omit the parameters $\xi_{0:3}$ to simplify the notations. Rearranging the rows and columns of the matrix on the left hand side gives the equation:

$$
\left[\begin{array}{ccc|ccc|ccc|c}
Q_0 & & I & & & -A_0^\top & & & & \\
& R_0 & & & & -B_0^\top & & & & \\
I & & & & & & & & & \\
\hline
& & & Q_1 & & I & & & -A_1^\top & \\
& & & & R_1 & & & & -B_1^\top & \\
-A_0 & -B_0 & & I & & & & & & \\
\hline
& & & & & & Q_2 & & I & -A_2^\top \\
& & & & & & & R_2 & & -B_2^\top \\
& & & -A_1 & -B_1 & & I & & & \\
\hline
& & & & & & -A_2 & -B_2 & &
\end{array}\right]
\left[\begin{array}{c} y_0\\v_0\\\eta_0\\y_1\\v_1\\\eta_1\\y_2\\v_2\\\eta_2\\\eta_3\end{array}\right]
=
\left[\begin{array}{c} Q_0\bar x_0\\0\\z_0\\Q_1\bar x_1\\0\\w_0\\Q_2\bar x_2\\0\\w_1\\w_2 - z_3\end{array}\right].
$$

Let $\hat{\Phi}_t^{t+k}$ denote the permutation matrix that permute $(y_t, v_t, y_{t+1}, \ldots, v_{t+k-1}; \eta_t, \ldots, \eta_{t+k})^\top$ to $(y_t, v_t, \eta_t, y_{t+1}, v_{t+1}, \eta_{t+1}, \ldots, y_{t+k-1}, v_{t+k-1}, \eta_{t+k-1}, \eta_{t+k})^\top$. We use $\hat{\Upsilon}_t^{t+k}(\xi_{t:t+k}) := (\hat{\Phi}_t^{t+k})\hat{H}_t^{t+k}(\xi_{t:t+k})(\hat{\Phi}_t^{t+k})^\top$ to denote the rearrangement of $\hat{H}_t^{t+k}(\xi_{t:t+k})$ as illustrated in the above equation, and use $\hat{\beta}_t^{t+k}(z, \xi_{t:t+k}) := (\hat{\Phi}_t^{t+k})\hat{b}_t^{t+k}(z, \xi_{t:t+k})$ to denote the corresponding rearrangement of $\hat{b}_t^{t+k}(z, \xi_{t:t+k})$.

Before showing the main result about the per-step error, we first show a technical lemma about the singular values of a block matrix in Lemma G.1.

**Lemma G.1.** *Consider a block matrix*

$$H = \begin{bmatrix} M & N^\top \\ N & 0 \end{bmatrix}.$$

*Here $M \in \mathbb{R}^{n_0 \times n_0}$ is a symmetric positive definite matrix that satisfies $\underline{\sigma}_M I \preceq M \le \overline{\sigma}_M I$ with $\underline{\sigma}_M > 0$, and $N \in \mathbb{R}^{n_1 \times n_0}$ with $n_1 \le n_0$ satisfies that $\underline{\sigma}_N \le \sigma(N) \le \overline{\sigma}_N$ with $\underline{\sigma}_N > 0$. Then $H$ satisfies that*

$$\min(\underline{\sigma}_M, 1) \cdot \overline{\sigma}_N \cdot \sqrt{\frac{\overline{\sigma}_M}{2\underline{\sigma}_M \overline{\sigma}_M + \underline{\sigma}_M (\underline{\sigma}_N)^2}} \le \sigma(H) \le \sqrt{2}(\overline{\sigma}_M + \overline{\sigma}_N).$$

*Proof of Lemma G.1.* We first establish the lower bound on $\sigma(H)$: Suppose the singular value decomposition of $NM^{-\frac{1}{2}}$ is given by

$$NM^{-\frac{1}{2}} = U\Sigma V^\top,$$

where $U \in \mathbb{R}^{n_1 \times n_1}$ and $V \in \mathbb{R}^{n_0 \times n_0}$ are unitary matrices and $\Sigma = diag(\sigma_1, \sigma_2, \ldots, \sigma_{n_1}) \in \mathbb{R}^{n_1 \times n_0}$ with $\frac{\overline{\sigma}_N}{\sqrt{\underline{\sigma}_M}} \ge \sigma_1 \ge \cdots \ge \sigma_m \ge \frac{\underline{\sigma}_N}{\sqrt{\overline{\sigma}_M}}$. We can decompose $H$ as

$$\begin{aligned} H &= \begin{bmatrix} M^{\frac{1}{2}} & 0 \\ 0 & I \end{bmatrix} \cdot \begin{bmatrix} I & M^{-\frac{1}{2}}F^\top \\ FM^{-\frac{1}{2}} & 0 \end{bmatrix} \cdot \begin{bmatrix} M^{\frac{1}{2}} & 0 \\ 0 & I \end{bmatrix} \\ &= \begin{bmatrix} M^{\frac{1}{2}} & 0 \\ 0 & I \end{bmatrix} \cdot \begin{bmatrix} V & 0 \\ 0 & U \end{bmatrix} \cdot \begin{bmatrix} I & \Sigma^\top \\ \Sigma & 0 \end{bmatrix} \cdot \begin{bmatrix} V^\top & 0 \\ 0 & U^\top \end{bmatrix} \cdot \begin{bmatrix} M^{\frac{1}{2}} & 0 \\ 0 & I \end{bmatrix}. \end{aligned} \qquad (25)$$

Note that for any $\alpha \in \mathbb{R}^{n_0}$ and $\beta \in \mathbb{R}^{n_1}$, we have

$$\begin{aligned} \left\| \begin{bmatrix} I & \Sigma^\top \\ \Sigma & 0 \end{bmatrix} \begin{bmatrix} \alpha \\ \beta \end{bmatrix} \right\|^2 &= \sum_{i=1}^{n_1} (\alpha_i + \sigma_i \beta_i)^2 + \sum_{i=n_1+1}^{n_0} \alpha_i^2 + \sum_{i=1}^{n_1} \sigma_i^2 \alpha_i^2 \\ &= \sum_{i=1}^{n_1} \left( \left(1 + \frac{\sigma_i^2}{2}\right) \left(\alpha_i + \frac{2\sigma_i}{2 + \sigma_i^2} \beta_i \right)^2 + \frac{\sigma_i^2}{2}\alpha_i^2 + \frac{\sigma_i^2}{2 + \sigma_i^2}\beta_i^2 \right) + \sum_{i=n_1+1}^{n_0} \alpha_i^2 \\ &\ge \left( \min_i \frac{\sigma_i^2}{2 + \sigma_i^2} \right) \left\| \begin{bmatrix} \alpha \\ \beta \end{bmatrix} \right\|^2 \\ &\ge \frac{\underline{\sigma}_M (\underline{\sigma}_N)^2}{2\underline{\sigma}_M \overline{\sigma}_M + \overline{\sigma}_M (\underline{\sigma}_N)^2} \left\| \begin{bmatrix} \alpha \\ \beta \end{bmatrix} \right\|^2. \end{aligned}$$

Therefore, by (25), we see that

$$\left\| H \begin{bmatrix} \alpha \\ \beta \end{bmatrix} \right\| \ge \min(\underline{\sigma}_M, 1) \cdot \underline{\sigma}_N \cdot \sqrt{\frac{\underline{\sigma}_M}{2\underline{\sigma}_M \overline{\sigma}_M + \overline{\sigma}_M (\underline{\sigma}_N)^2}} \cdot \left\| \begin{bmatrix} \alpha \\ \beta \end{bmatrix} \right\|.$$

This finishes the proof of the lower bound.

For the upper bound, note that

$$\begin{aligned} \left\| H \begin{bmatrix} \alpha \\ \beta \end{bmatrix} \right\| &\le \|M\alpha + N^\top \beta\| + \|N\alpha\| \\ &\le \|M\alpha\| + \|N^\top \beta\| + \|N\alpha\| \end{aligned}$$

$$\leq (\overline{\sigma}_M + \overline{\sigma}_N)(\|x\| + \|y\|)$$
$$\leq \sqrt{2}(\overline{\sigma}_M + \overline{\sigma}_N)\left\|\begin{bmatrix}\alpha\\\beta\end{bmatrix}\right\|.$$

$\square$

Since the primal-dual optimal solution to $\iota_t^T(z, \xi_{t:T}; F_T)$ and $\iota_t^{t+k}(z, \xi_{t:t+k}; \mathbb{I})$ are given by $\left(\Upsilon_t^T(\xi_{t:T})\right)^{-1}\beta_t^T(z, \xi_{t:T})$ and $\left(\hat{\Upsilon}_t^{t+k}(\xi_{t:t+k})\right)^{-1}\hat{\beta}_t^{t+k}(z, \xi_{t:t+k})$ respectively, it is critical to establish the exponentially decaying bounds for the matrices $\left(\Upsilon_t^T(\xi_{t:T})\right)^{-1}$ and $\left(\hat{\Upsilon}_t^{t+k}(\xi_{t:t+k})\right)^{-1}$. Note that after the rearrangement, $\Upsilon_t^T(\xi_{t:T})$ is a block matrix with $(T - t + 1) \times (T - t + 1)$ blocks, indexed by $(i, j) \in [t, T]^2$; $\Upsilon_t^{t+k}(\xi_{t:t+k})$ is a block matrix with $(k + 1) \times (k + 1)$ blocks, indexed by $(i, j) \in [t, t + k]^2$.

**Lemma G.2.** *Under Assumption 4.2, the following inequalities hold for the norm of the block entries of $\left(\Upsilon_t^T(\xi_{t:T})\right)^{-1}$ and $\left(\hat{\Upsilon}_t^{t+k}(\xi_{t:t+k})\right)^{-1}$:*

$$\left\|\left(\Upsilon_t^T(\xi_{t:T})^{-1}\right)_{ij}\right\| \leq C_2\lambda_2^{|i-j|}, \forall (i,j) \in [t, T]^2, \forall \xi_{t:T} \in \Xi_{t:T},$$
$$\left\|\left(\hat{\Upsilon}_t^{t+k}(\xi_{t:t+k})^{-1}\right)_{ij}\right\| \leq C_2\lambda_2^{|i-j|}, \forall (i,j) \in [t, t+k]^2, \forall \xi_{t:t+k} \in \Xi_{t:t+k}. \tag{26}$$

*Further, the following inequalities hold for the norm of differences between the block entries of $\left(\Upsilon_t^T(\xi_{t:T})\right)^{-1}$ and $\left(\hat{\Upsilon}_t^{t+k}(\xi_{t:t+k})\right)^{-1}$: For all $(i,j) \in [t, T]^2$ and $\xi_{t:T} \in \Xi_{t:T}$, we have*

$$\left\|\left(\Upsilon_t^T(\xi_{t:T})^{-1} - \Upsilon_t^T(\xi'_{t:T})^{-1}\right)_{ij}\right\| \leq C'_2\sum_{\tau=t}^{T}\lambda_2^{|\tau-i|+|\tau-j|} \cdot \|\xi_\tau - \xi'_\tau\|, \tag{27}$$

*For all $(i,j) \in [t, t+k]^2$ and $\xi_{t:T} \in \Xi_{t:t+k}$ with $t < T - k$, we have*

$$\left\|\left(\hat{\Upsilon}_t^{t+k}(\xi_{t:t+k})^{-1} - \hat{\Upsilon}_t^{t+k}(\xi'_{t:t+k})^{-1}\right)_{ij}\right\| \leq C'_2\sum_{\tau=t}^{t+k}\lambda_2^{|\tau-i|+|\tau-j|} \cdot \|\xi_\tau - \xi'_\tau\|, \tag{28}$$

*where the constants $C_2, C'_2$, and $\lambda_2$ are given by*

$$\lambda_2 = \left(\frac{\overline{\sigma}_H - \underline{\sigma}_H}{\overline{\sigma}_H + \underline{\sigma}_H}\right)^{\frac{1}{2}}, C_2 = \frac{4(\ell + 1 + a + b)}{\underline{\sigma}_H^2 \cdot \lambda_2},$$
$$C'_2 = C_2^2\left(\max\{L_Q + L_R, L_P\} + \frac{2}{\lambda_2}(L_A + L_B)\right).$$

*where $\underline{\sigma}_H$ and $\overline{\sigma}_H$ are defined as*

$$\underline{\sigma}_H := \min(\mu, 1) \cdot (a + b + 1) \cdot \sqrt{\frac{\ell}{2\mu\ell + \mu\sigma^2}}, \text{ and } \overline{\sigma}_H := \sqrt{2}(\ell + a + b + 1).$$

*Proof of Lemma G.2.* In the proof, we only show the results for $\Upsilon_t^T$. The results for $\hat{\Upsilon}_t^{t+k}$ can be shown using the same method.

We first show (26) holds. By Lemma G.1, we know that Note that $\Upsilon_t^T(\xi_{t:T})^2$ is a positive definite matrix that has band width $4$ and satisfies

$$\underline{\sigma}_H^2 I \preceq \Upsilon_t^T(\xi_{t:T})^2 \preceq \overline{\sigma}_H^2 I.$$

Using the same method as the proof of Lemma B.1 in [35], one can show that for any $(i,j) \in [t, T]^2$,

$$\left(\left(\Upsilon_t^T(\xi_{t:T})^2\right)^{-1}\right)_{ij} \leq \frac{2}{\underline{\sigma}_H^2} \cdot \lambda_2^{|i-j|}. \tag{29}$$

Note that $\Upsilon_t^T(\xi_{t:T})^{-1} := \Upsilon_t^T(\xi_{t:T}) \cdot \left(\Upsilon_t^T(\xi_{t:T})^2\right)^{-1}$. Thus we see that

$$\left(\Upsilon_t^T(\xi_{t:T})^{-1}\right)_{ij} = \left(\Upsilon_t^T(\xi_{t:T}) \cdot \left(\Upsilon_t^T(\xi_{t:T})^2\right)^{-1}\right)_{ij}$$

$$= \sum_{k=t}^{T} \Upsilon_t^T(\xi_{t:T})_{ik} \cdot \left(\left(\Upsilon_t^T(\xi_{t:T})^2\right)^{-1}\right)_{kj}$$

$$= \sum_{k=i-1}^{i+1} \Upsilon_t^T(\xi_{t:T})_{ik} \cdot \left(\left(\Upsilon_t^T(\xi_{t:T})^2\right)^{-1}\right)_{kj}.$$

Therefore, by (29), we see that

$$\left\|\left(\Upsilon_t^T(\xi_{t:T})^{-1}\right)_{ij}\right\| \leq \frac{4(\ell+1+a+b)}{\underline{\sigma}_H^2 \cdot \lambda_2} \cdot \lambda_2^{|i-j|}, \forall (i,j) \in [t,T]^2. \tag{30}$$

Note that we have

$$\Upsilon_t^T(\xi_{t:T})^{-1} - \Upsilon_t^T(\xi'_{t:T})^{-1} = -(\Upsilon_t^T(\xi'_{t:T}))^{-1} \left(\Upsilon_t^T(\xi_{t:T}) - \Upsilon_t^T(\xi'_{t:T})\right) \Upsilon_t^T(\xi_{t:T})^{-1}. \tag{31}$$

To simplify the notation, we define

$$E_\tau := \left(\Upsilon_t^T(\xi_{t:T}) - \Upsilon_t^T(\xi'_{t:T})\right)_{\tau\tau}, \forall \tau \in [t,T],$$

and

$$E'_\tau := \left(\Upsilon_t^T(\xi_{t:T}) - \Upsilon_t^T(\xi'_{t:T})\right)_{(\tau+1)\tau}, \forall \tau \in [t,T-1].$$

The right hand side of (31) is a linear equation. Thus we can study the following 3 equations separately and sum them up:

$$\Phi_\tau := (\Upsilon_t^T(\xi'_{t:T}))^{-1} \begin{bmatrix} 0 & & & & & & \\ & \ddots & & & & & \\ & & 0 & & & & \\ & & & E_\tau & & & \\ & & & & 0 & & \\ & & & & & \ddots & \\ & & & & & & 0 \end{bmatrix} (\Upsilon_t^T(\xi_{t:T}))^{-1}, \forall \tau \in [t,T], \tag{32a}$$

$$\Phi_\tau^L := (\Upsilon_t^T(\xi'_{t:T}))^{-1} \begin{bmatrix} 0 & & & & & \\ & \ddots & & & & \\ & & 0 & & & \\ & & E'_\tau & 0 & & \\ & & & & \ddots & \\ & & & & & 0 \end{bmatrix} (\Upsilon_t^T(\xi_{t:T}))^{-1}, \forall \tau \in [t,T-1], \tag{32b}$$

$$\Phi_\tau^U := (\Upsilon_t^T(\xi'_{t:T}))^{-1} \begin{bmatrix} 0 & & & & & \\ & \ddots & & & & \\ & & 0 & (E'_\tau)^\top & & \\ & & & 0 & & \\ & & & & \ddots & \\ & & & & & 0 \end{bmatrix} (\Upsilon_t^T(\xi_{t:T}))^{-1}, \forall \tau \in [t,T-1]. \tag{32c}$$

By (32a), (32b), and (32c), we see that

$$\|(\Phi_\tau)_{ij}\| \leq \left\|\left((\Upsilon_t^T(\xi'_{t:T}))^{-1}\right)_{i\tau}\right\| \cdot \|E_\tau\| \cdot \left\|\left(\Upsilon_t^T(\xi_{t:T})^{-1}\right)_{\tau j}\right\|$$

$$\leq C_2^2 \cdot \lambda_2^{|\tau-i|+|\tau-j|} \cdot \|E_\tau\|, \tag{33a}$$

$$\|(\Phi_\tau^L)_{ij}\| \leq \left\|\left((\Upsilon_t^T(\xi'_{t:T}))^{-1}\right)_{i\tau}\right\| \cdot \|E'_\tau\| \cdot \left\|\left(\Upsilon_t^T(\xi_{t:T})^{-1}\right)_{(\tau+1)j}\right\|$$

$$\leq C_2^2 \cdot \lambda_2^{|\tau-i|+|\tau+1-j|} \cdot \|E'_\tau\|, \tag{33b}$$

$$\left\| (\Phi_\tau^U)_{ij} \right\| \le \left\| \left( (\Upsilon_t^T (\xi'_{t:T}))^{-1} \right)_{i(\tau+1)} \right\| \cdot \| E'_\tau \| \cdot \left\| \left( \Upsilon_t^T (\xi_{t:T})^{-1} \right)_{\tau j} \right\|$$

$$\le C_2^2 \cdot \lambda_2^{|\tau+1-i|+|\tau-j|} \cdot \| E'_\tau \|, \tag{33c}$$

where we use the bound (30) on the norm of individual block entries in (33a), (33b), and (33c). Summing these inequalities up over $\tau$, we see that

$$\left\| \left( \Upsilon_t^T (\xi_{t:T})^{-1} - \Upsilon_t^T (\xi'_{t:T})^{-1} \right)_{ij} \right\|$$

$$= \left\| \sum_{\tau=t}^{T} (\Phi_\tau)_{ij} + \sum_{\tau=t}^{T-1} (\Phi_\tau^L)_{ij} + \sum_{\tau=t}^{T-1} (\Phi_\tau^U)_{ij} \right\| \tag{34a}$$

$$\le \sum_{\tau=t}^{T} \| (\Phi_\tau)_{ij} \| + \sum_{\tau=t}^{T-1} \| (\Phi_\tau^L)_{ij} \| + \sum_{\tau=t}^{T-1} \| (\Phi_\tau^U)_{ij} \| \tag{34b}$$

$$\le C_2^2 \left( \sum_{\tau=t}^{T} \lambda_2^{|\tau-i|+|\tau-j|} \cdot \| E_\tau \| + \frac{2}{\lambda_2} \sum_{\tau=t}^{T-1} \lambda_2^{|\tau-i|+|\tau-j|} \cdot \| E'_\tau \| \right) \tag{34c}$$

$$\le C_2^2 \left( \max\{L_Q + L_R, L_P\} + \frac{2}{\lambda_2} (L_A + L_B) \right) \sum_{\tau=t}^{T} \lambda_2^{|\tau-i|+|\tau-j|} \cdot \| \xi_\tau - \xi'_\tau \|, \tag{34d}$$

where we use (31) in (34a); we use the triangle inequality in (34b); we use (33) in (34c); we use the Lipschitzness of dynamical and cost matrices in $\xi$ (Assumption 4.2) in (34d). $\qquad \square$

With Lemma G.2, we can derive the perturbation bounds specified by Property 3.1.

**Theorem G.3.** *Under Assumption 4.2, Property 3.1 holds for arbitrary positive constant $R$ and $q_1(t) = H_2 \lambda_2^{2t}$, $q_2(t) = H_2 \lambda_2^t$, and $q_3(t) = H_2 \lambda_2^t$, where $\lambda_2$ is defined in Lemma G.2, and $H_2$ is given by*

$$H_2 = C_2' \left( \frac{2(\ell D_{\bar{x}} + D_w)}{1 - \lambda_2} + R + D_{x^*} + 1 \right) + C_2 \left( L_w + \ell L_{\bar{x}} + D_{\bar{x}} L_Q + 1 \right).$$

*Proof of Theorem G.3.* For $t < T - k$, under the specification of Property 3.1, we see that

$$\left\| \psi_t^{t+k} (z, \xi_{t:t+k}; \mathbb{I})_{v_t} - \psi_t^{t+k} (z, \xi'_{t:t+k}; \mathbb{I})_{v_t} \right\|$$

$$\le \left\| \left( \hat{\Upsilon}_t^{t+k} (\xi_{t:t+k})^{-1} \hat{\beta}_t^{t+k} (z, \xi_{t:t+k}) - \hat{\Upsilon}_t^{t+k} (\xi'_{t:t+k})^{-1} \hat{\beta}_t^{t+k} (z, \xi'_{t:t+k}) \right)_{v_t} \right\| \tag{35a}$$

$$\le \left\| \left( \left( \hat{\Upsilon}_t^{t+k} (\xi_{t:t+k})^{-1} - \hat{\Upsilon}_t^{t+k} (\xi'_{t:t+k})^{-1} \right) \hat{\beta}_t^{t+k} (z, \xi_{t:t+k}) \right)_{v_t} \right\|$$

$$+ \left\| \left( \hat{\Upsilon}_t^{t+k} (\xi'_{t:t+k})^{-1} \left( \hat{\beta}_t^{t+k} (z, \xi_{t:t+k}) - \hat{\beta}_t^{t+k} (z, \xi'_{t:t+k}) \right) \right)_{v_t} \right\|, \tag{35b}$$

where we used the KKT condition in (35a) and the triangle inequality in (35b).

For the first term in (35b), we see that

$$\left\| \left( \left( \hat{\Upsilon}_t^{t+k} (\xi_{t:t+k})^{-1} - \hat{\Upsilon}_t^{t+k} (\xi'_{t:t+k})^{-1} \right) \cdot \hat{\beta}_t^{t+k} (z, \xi_{t:t+k}) \right)_{v_t} \right\|$$

$$\le \sum_{\tau=t}^{t+k} \left\| \left( \hat{\Upsilon}_t^{t+k} (\xi_{t:t+k})^{-1} - \hat{\Upsilon}_t^{t+k} (\xi'_{t:t+k})^{-1} \right)_{t\tau} \right\| \cdot \left\| \hat{\beta}_t^{t+k} (z, \xi_{t:t+k})_\tau \right\| \tag{36a}$$

$$\le C_2' \left( \sum_{\tau=0}^{k} \lambda_2^{2\tau} \| \xi_{t+\tau} - \xi'_{t+\tau} \| \right) \cdot \| z \| + \sum_{\tau=t}^{t+k} C_2' \left( \sum_{i=t}^{t+k} \lambda_2^{i-t+|i-\tau|} \| \xi_i - \xi'_i \| \right) \cdot (\ell D_{\bar{x}} + D_w)$$

$$+ C_2' \lambda_2^k \cdot \left( \sum_{\tau=0}^{k} \| \xi_{t+\tau} - \xi'_{t+\tau} \| \right) \cdot \| \xi_{t+k} \| \tag{36b}$$

$$\leq C_2' \sum_{\tau=0}^{k} \lambda_2^{2\tau} \delta_{t+\tau} \cdot \|z\| + C_2' \left( \frac{2(\ell D_{\bar{x}} + D_w)}{1 - \lambda_2} + R + D_{x^*} \right) \sum_{\tau=0}^{k} \lambda_2^{\tau} \delta_{t+\tau}, \tag{36c}$$

where we use the triangle inequality in (36a); we use Lemma G.2 and the bounds on each entry of $\hat{\beta}_t^{t+k}(z, \xi_{t:t+k})$ in (36b); we rearrange the terms and use $\xi_{t+k} \in \mathcal{B}(x_{t+k}^*, R)$ in (36c). For the second error term (35b), we see that

$$\left\| \left( \hat{\Upsilon}_t^{t+k}(\xi_{t:t+k}')^{-1} \left( \hat{\beta}_t^{t+k}(z, \xi_{t:t+k}) - \hat{\beta}_t^{t+k}(z, \xi_{t:t+k}') \right) \right)_{v_t} \right\|$$

$$\leq C_2 \sum_{\tau=t}^{t+k} \lambda_2^{\tau-t}(L_w + \ell L_{\bar{x}} + D_{\bar{x}} L_Q)\delta_\tau + C_2 \lambda_2^k \delta_{t+k}, \tag{37}$$

where we use the following inequality to bound the difference between $\hat{\beta}_t^{t+k}(z, \xi_{t:t+k})$ and $\hat{\beta}_t^{t+k}(z, \xi_{t:t+k}')$:

$$\|Q_\tau(\xi_\tau)\bar{x}_\tau(\xi_\tau) - Q_\tau(\xi_\tau')\bar{x}_\tau(\xi_\tau')\|$$
$$\leq \|Q_\tau(\xi_\tau)\bar{x}_\tau(\xi_\tau) - Q_\tau(\xi_\tau')\bar{x}_\tau(\xi_\tau)\| + \|Q_\tau(\xi_\tau')\bar{x}_\tau(\xi_\tau) - Q_\tau(\xi_\tau')\bar{x}_\tau(\xi_\tau')\|$$
$$\leq \|Q_\tau(\xi_\tau) - Q_\tau(\xi_\tau')\| \cdot \|\bar{x}_\tau(\xi_\tau)\| + \|Q_\tau(\xi_\tau')\| \cdot \|\bar{x}_\tau(\xi_\tau) - \bar{x}_\tau(\xi_\tau')\|$$
$$\leq (L_Q D_{\bar{x}} + \ell L_{\bar{x}})\delta_\tau.$$

Substituting (36) and (37) into (35) gives that for any $t < T - k$,

$$\left\| \psi_t^{t+k}\left(z, \xi_{t:t+k}; \mathbb{I}\right)_{v_t} - \psi_t^{t+k}\left(z, \xi_{t:t+k}'; \mathbb{I}\right)_{v_t} \right\| \leq \left( \sum_{\tau=0}^{k} q_1(\tau)\delta_{t+\tau} \right) \|z\| + \sum_{\tau=0}^{k} q_2(\tau)\delta_{t+\tau}$$

under the specification that $\xi_{t:t+k-1} \in \Xi_{t:t+k-1}, \xi_{t:t+k-1}' = \xi_{t:t+k-1}^*; \xi_{t+k}, \xi_{t+k}' \in \mathcal{B}(x_{t+k}^*, R)$. We can use a similar methods to show that for any $t \geq T - k$,

$$\left\| \psi_t^T\left(z, \xi_{t:T}; F_T\right)_{v_t} - \psi_t^T\left(z, \xi_{t:T}'; F_T\right)_{v_t} \right\| \leq \left( \sum_{\tau=0}^{T-t} q_1(\tau)\delta_{t+\tau} \right) \|z\| + \sum_{\tau=0}^{T-t} q_2(\tau)\delta_{t+\tau}$$

under the specification that $\xi_{t:T} \in \Xi_{t:T}, \xi_{t:T}' = \xi_{t:T}^*$.

For any $t < T$, we see that

$$\left\| \psi_t^T(z, \xi_{t:T}^*; F_T)_{y_\tau/v_\tau} - \psi_t^T(z', \xi_{t:T}^*; F_T)_{y_\tau/v_\tau} \right\|$$
$$\leq \left\| \left( \Upsilon_t^T(\xi_{t:T}^*)^{-1} \left( \beta_t^T(z, \xi_{t:T}^*) - \beta_t^T(z', \xi_{t:T}^*) \right) \right)_{y_\tau/v_\tau} \right\| \tag{38a}$$
$$\leq \left\| \left( \Upsilon_t^T(\xi_{t:T}^*)^{-1} \right)_{\tau t} \right\| \cdot \|z - z'\|$$
$$\leq C_2 \lambda_2^{\tau-t} \|z - z'\|, \tag{38b}$$

where we use the KKT condition in (38a); we use Lemma G.2 in (38b). □

Now we come back to the proof of Theorem 4.2.

*Proof of Theorem 4.2.* By Theorem G.3, Property 3.1 holds for arbitrary positive constant $R$ and $q_1(t) = H_2 \lambda_2^{2t}, q_2(t) = H_2 \lambda_2^t$, and $q_3(t) = H_2 \lambda_2^t$, where the decay rate $\lambda_2 \in (0, 1)$ and constant $H_2$ depends on $R$. We set $R := D_{x^*} + D_{\bar{x}}$ so that $\mathsf{MPC}_k$ with terminal state $\bar{x}_{t+k}(\xi_{t+k|t})$ satisfies the assumption of Theorem 3.3. The constant $H_2$ is given by

$$H_2 = C_2' \left( \frac{2(\ell D_{\bar{x}} + D_w)}{1 - \lambda_2} + 2D_{x^*} + D_{\bar{x}} + 1 \right) + C_2 \left( L_w + \ell L_{\bar{x}} + D_{\bar{x}} L_Q + 1 \right).$$

By Theorem 3.3, in order to achieve the claimed dynamic regret bound in Theorem 4.2, a sufficient condition is that the prediction errors $\rho_{t,\tau}$ satisfy

$$\sum_{\tau=0}^{k} \lambda_2^\tau \rho_{t,\tau} \leq \frac{(1 - \lambda_2)^2 (D_{x^*} + D_{\bar{x}})}{2H_2^2 L_g \left( (1 - \lambda_2)(D_{x^*} + D_{\bar{x}}) + H_2(D_{x^*} + 1) \right)},$$

and the prediction horizon $k$ satisfies that

$$\lambda_2^k \leq \frac{(1-\lambda_2)^2}{4H_2^2 L_g\left((1-\lambda_2)(D_{x^*}+D_{\bar{x}})+H_2(D_{x^*}+1)\right)}.$$

$\square$

We provide two example systems where the dynamics is unknown but the uniform controllability assumption (Assumption G.1) is satisfied for all possible uncertainty parameters.

**Example G.4** (Inverted pendulum with unknown mass). *We consider the linearized inverted pendulum dynamics in discrete state space. The dynamics of the system is given by*

$$\begin{bmatrix} x_{t+1} \\ \dot{x}_{t+1} \\ \phi_{t+1} \\ \dot{\phi}_{t+1} \end{bmatrix} = \underbrace{\begin{bmatrix} 1 & \delta & 0 & 0 \\ 0 & 1+\frac{-(I+ml^2)b\delta}{I(M+m)+Mml^2} & \frac{m^2gl^2\delta}{I(M+m)+Mml^2} & 0 \\ 0 & 0 & 1 & \delta \\ 0 & \frac{-mlb\delta}{I(M+m)+Mml^2} & \frac{mgl(M+m)\delta}{I(M+m)+Mml^2} & 1 \end{bmatrix}}_{A(M)} \begin{bmatrix} x_t \\ \dot{x}_t \\ \phi_t \\ \dot{\phi}_t \end{bmatrix} + \underbrace{\begin{bmatrix} 0 \\ \frac{(I+ml^2)\delta}{I(M+m)+Mml^2} \\ 0 \\ \frac{ml\delta}{I(M+m)+Mml^2} \end{bmatrix}}_{B(M)} u_t,$$

*where $M$ is the mass of the cart, $m$ is the mass of the pendulum, $b$ is the coefficient of friction for cart, $l$ is the length to the center of the mass of the pendulum, $I$ is the mass moment of inertia of the pendulum, $x_t$ is the position of the cart, $\phi_y$ is the angle of the pendulum, and $\delta$ is step size of discretization. For simplicity, we assume all system parameters are known except the mass of the cart $M$, which is an unknown value in the interval $\left[\underline{M}, \overline{M}\right]$ for some constants $\overline{M} > \underline{M} > 0$. The controllability matrix of the dynamical system satisfies that*

$$\det\left[B(M), A(M)B(M), A(M)^2B(M), A^3(M)B(M)\right] = \frac{\delta^{10}g^2l^4m^4}{(IM+m(I+l^2M))^4}$$

$$\geq \frac{\delta^{10}g^2l^4m^4}{(I\overline{M}+m(I+l^2\overline{M}))^4} > 0.$$

*Therefore, we can use Theorem 1 in [56] to show*

$$\sigma_{min}\left[B(M), A(M)B(M), A(M)^2B(M), A^3(M)B(M)\right] \geq \sigma_0$$

*for some positive constant $\sigma_0$, which implies Assumption G.1 by Lemma 12 in [37].*

**Example G.5** (Frequency regulation with unknown inertia). *We consider the power grid dynamics with $n$ nodes studied in [18]:*

$$\underbrace{\begin{bmatrix} \dot{\theta} \\ \dot{\omega} \end{bmatrix}}_{\dot{x}(t)} = \underbrace{\begin{bmatrix} 0 & I \\ -M_{q(t)}^{-1}L & -M_{q(t)}^{-1}D \end{bmatrix}}_{\hat{A}(t)} \underbrace{\begin{bmatrix} \theta \\ \omega \end{bmatrix}}_{x(t)} + \underbrace{\begin{bmatrix} 0 \\ M_{q(t)}^{-1} \end{bmatrix}}_{\hat{B}(t)} \underbrace{p_{in}}_{u(t)}.$$

*Here, $\theta, \omega \in \mathbb{R}^n$ are the vectors of voltage phase angles and frequencies. The state $x(t)$ is a stacked vector of $\theta$ and $\omega$, and $u(t) = p_{in}$ is the power input. $D$ is a diagonal matrix whose entries represent droop control coefficients, and $L$ is the Laplacian matrix of the network of nodes. $M_{q(t)}$ is the inertia matrix in mode $q(t) \in \{1, \ldots, m\}$ and is time-varying. It is further assumed to be in the form $M_{q(t)} = m_{q(t)}I$, where $m_{q(t)}$ is a scalar that represents the inertia coefficient at time $t$. We assume all system parameters are known except the inertia coefficient, and we define $\xi_t := m_{q(t)}$.*

*For simplicity, we use a different discretization technique with [18] so that it is easier to verify uniform controllability. Specifically, we write the discrete-time system as*

$$x_{t+1} = \underbrace{\left(I+\delta\hat{A}(t)\right)}_{A(\xi_t)} x_t + \underbrace{\delta\hat{B}(t)}_{B(\xi_t)} u_t,$$

*where $\delta$ is the step size of discretization. We see that when $0 < \underline{m} \leq m_{q(t)} \leq \overline{m} < +\infty$ holds for some positive constants $\underline{m}, \overline{m}$ for all possible inertia coefficient, we have*

$$\left|\det\left[B(\xi_{t+1}), A(\xi_{t+1})B(\xi_t)\right]\right| \geq \frac{\delta^{3n}}{\overline{m}^{2n}} > 0.$$

*Thus, we can use Theorem 1 in [56] to show that*

$$\sigma_{min}\left[B(\xi_{t+1}), A(\xi_{t+1})B(\xi_t)\right] \geq \sigma_0$$

*holds for all possible inertia coefficient at all time steps for some positive constant $\sigma_0$, which implies Assumption G.1 by Lemma 12 in [37].*

# H  Assumptions and Proofs of Section 5

To introduce the SSOSC assumption, we first define the *reduced Hessian* of the Lagrangian.

**Definition H.1** (reduced Hessian). *For a constrained optimization problem with primal variable $z$ and dual variable $\eta$, let $H = \nabla^2_{zz}\mathcal{L}$ denote the Hessian of the Lagrangian $\mathcal{L}(z, \eta; \xi)$. Let $G$ denote the* **active constraints Jacobian**, *i.e. Jacobian of all equality constraints and active inequality constraints, and let $Z$ be the null-space matrix of $G$ (i.e., the column vectors of $Z$ form an orthonormal basis of the null space of $G$). Then the* **reduced Hessian** *is defined as $H_{\mathrm{re}}(z, \eta; \xi) := Z^\top H Z$.*

We define the concept of *singular spectrum bounds* for a specific instance of FTOCP:

**Definition H.2** (singular spectrum bounds). *Consider the FTOCP $\iota^{t_2}_{t_1}(z, \xi_{t_1:t_2}; F)$. The positive real numbers $\overline{\sigma}_H, \overline{\sigma}_R, \underline{\sigma}_H$ are called* **singular spectrum bounds** *for this specific instance of FTOCP[2] if they satisfy that*

$$\overline{\sigma}_H \geq \overline{\overline{\sigma}}_H(z, \xi_{t_1:t_2}), \overline{\sigma}_R \geq \overline{\overline{\sigma}}_R(z, \xi_{t_1:t_2}), \text{ and } 0 < \underline{\sigma}_H \leq \underline{\underline{\sigma}}_H(z, \xi_{t_1:t_2}),$$

*where $\overline{\overline{\sigma}}_H, \overline{\overline{\sigma}}_R$, and $\underline{\underline{\sigma}}_H$ are defined in (4.16a-c) in [50].*

**Assumption H.1.** *We make the following assumptions on the costs, dynamics, and constraints of an FTOCP $\iota^{t_2}_{t_1}(z, \xi_{t_1:t_2}; F)$:*

1. *All cost functions, dynamical functions, and constraint functions are twice continuously differentiable in $(x_t, u_t)$ and $\xi_t$ [3].*

2. *(SSOSC) The reduced Hessian at the optimal primal-dual solution is positive-definite.*

3. *(LICQ) The active constraints Jacobian $G$ at the optimal primal-dual solution has full row rank, i.e. $\sigma_{\min}(G) > 0$.*

4. *(Uniform singular spectrum bounds) There exist positive singular spectrum bounds $\overline{\sigma}_H, \overline{\sigma}_R, \underline{\sigma}_H$ for all FTOCP specifications below:*

   (a) *$t_1 = t, t_2 = t + k$ for $t < T - k$:*

   $$z \in \mathcal{B}(x^*_t, R), \xi_{t:t+k-1} \in \Xi_{t:t+k-1}, \xi_{t+k} \in \mathcal{B}(x^*_{t+k}, R), F = \mathbb{I}.$$

   (b) *$t_1 = t, t_2 = T$ for $t < T$:*

   $$z \in \mathcal{B}(x^*_t, R), \xi_{t:T} \in \Xi_{t:T}, F = F_T.$$

We remind the readers that Lemma 12 in [37] shows that Lipshitzness of dynamics and uniform controllability together imply uniform LICQ property of the system.

Under Assumption H.1, we know Property 3.1 holds for $q_1(t) = 0$, $q_2(t) = H_3\lambda^t_3$, and $q_3(t) = H_3\lambda^t_3$ for some $H_3 > 0$ and $\lambda_3 \in (0, 1)$ by Theorem 4.5 in [50].

**Theorem H.1.** *Under Assumption H.1 that holds for some $R > 0$, Property 3.1 holds for $q_1(t) = 0$, $q_2(t) = H_3\lambda^t_3$, and $q_3(t) = H_3\lambda^t_3$ with the same $R$. The coefficient $H_3$ and decay factor $\lambda_3$ are given by*

$$H_3 := \left(\frac{\overline{\sigma}_H\overline{\sigma}_R}{\underline{\sigma}^2_H}\right)^{\frac{1}{2}}, \text{ and } \lambda_3 := \left(\frac{\overline{\sigma}^2_H - \underline{\sigma}^2_H}{\overline{\sigma}^2_H + \underline{\sigma}^2_H}\right)^{\frac{1}{8}}.$$

Combining Theorem H.1 with the Pipeline Theorem (Theorem 3.3) finishes the proof of Theorem 5.1. Note that when $\xi^*_t \leq \frac{(1-\lambda_3)R}{H_3}$, we know that $\left\|\psi^T_0(x_0, \mathbf{0}; F_T)_{y_{t+k}} - x^*_{t+k}\right\| \leq R$. Thus using $\psi^T_0(x_0, \mathbf{0}; F_T)_{y_{t+k}}$ as the terminal state of $\mathsf{MPC}_k$ at time step $t$ can satisfy the requirement of Theorem 5.1.

---

[2] We remind the reader that the functions $\overline{\overline{\sigma}}_H, \overline{\overline{\sigma}}_R$, and $\underline{\underline{\sigma}}_H$ depend on the form of FTOCP, i.e., different horizon $[t_1, t_2]$ and different terminal cost function $F$.

[3] If the terminal function $F$ is an indicator function of some state, we view it as a constraint instead of cost.

# I Inventory control with constraints

In this section, we present two examples about the perturbation bounds under a simple inventory control dynamics. In the first example, we use the results in [50] to show a general exponentially decaying perturbation bound holds when we only have one-sided constraints on control input $u_t$.

**Theorem I.1.** *Consider the optimal control problem where the state $x_t$ and the control input $u_t$ are both in $\mathbb{R}$. The dynamics and constraints are given by*

$$x_{t+1} = x_t + u_t, \ s.t. \ x_t \in [-1, 1], u_t \geq -\frac{4}{5}$$

*for all $t$. The stage cost is given by $f_t(x_t, u_t; \xi_t)$, where $f_t$ is convex and $\ell$-smooth. We also assume that $f_t$ is $\mu$-strongly convex in its first variable $x_t$. Then, for any positive integer $p \geq 3$, we have*

$$\left| \psi_0^p \left( x_0, \xi_{0:p-1}, \zeta_p; \mathbb{I} \right)_{x_h} - \psi_0^p \left( x_0', \xi_{0:p-1}', \zeta_p'; \mathbb{I} \right)_{x_h} \right|$$

$$\leq C \left( \lambda^h |x_0 - x_0'| + \sum_{\tau=0}^{p-1} \lambda^{|h-\tau|} |\xi_\tau - \xi_\tau'| + \lambda^{p-h} |\zeta_p - \zeta_p'| \right), \tag{39}$$

*for all $x_0, \zeta_p \in [-1, 1]$, where $h \in \{1, \ldots, p\}$ and $C > 0, \lambda \in (0, 1)$ are some constants.*

*Proof of Theorem I.1.* We can rewrite the optimization problem to remove the equality constraints as following:

$$\min_{x_{1:p-1}} \sum_{t=0}^{p-1} f_t(x_t, x_{t+1} - x_t; \xi_t) \tag{40a}$$

$$\text{s.t. } -1 \leq x_t \leq 1, \forall t \in \{1, 2, \ldots, p-1\}, \tag{40b}$$

$$x_t - x_{t-1} \geq -\frac{4}{5}, \forall t \in \{1, 2, \ldots, p\}, \tag{40c}$$

where $x_p = \zeta_p$.

Note that for any time index $t \in \{1, 2, \ldots, p-1\}$, at most 2 constraints that involves $x_t$ can be active. They can be chosen from the 4 possible constraints that involves $x_t$:

$$x_t \geq -1, x_t \leq 1, x_t - x_{t-1} \geq -\frac{4}{5}, x_{t+1} - x_t \geq -\frac{4}{5}.$$

And for any time index $t \in \{1, 2, \ldots, p-2\}$, the 3 consecutive "coupling" constraints

$$x_t - x_{t-1} \geq -\frac{4}{5}, x_{t+1} - x_t \geq -\frac{4}{5}, x_{t+2} - x_{t+1} \geq -\frac{4}{5}, \tag{41}$$

cannot activate simultaneously. Let $\sigma_0$ denote the smallest singular value of matrix

$$\begin{bmatrix} 1 & & \\ -1 & 1 & \\ & -1 & 1 \end{bmatrix}.$$

Therefore, in the context of Theorem 4.5 in [50], we see that $\underline{\sigma}\left(\nabla_{xy}\mathcal{L}\left(z^\dagger(\xi); \xi\right)[\mathcal{B}, \mathcal{B}]\right)$ is lower bounded by $\sigma_0$ and upper bounded by 2. Since we also have that

$$\mu I \preceq \nabla_{xx}\mathcal{L}\left(z^\dagger(\xi); \xi\right)[\mathcal{B}, \mathcal{B}] \preceq 5\ell I.$$

By Lemma G.1, we further see that we can set $\underline{\sigma}_H$ and $\overline{\sigma}_H$ as

$$\underline{\sigma}_H := 2\min(\mu, 1)\sqrt{\frac{5\ell}{10\mu\ell + \mu\sigma_0^2}}, \overline{\sigma}_H := \sqrt{2}(5\ell + 2).$$

We can set $\overline{\sigma}_R := \ell$. Applying Theorem 4.5 in [50] finishes the proof. $\square$

As a remark, we have already certified Assumption H.1 in the proof of Theorem I.1 and the perturbation bound provided by Theorem I.1 is more general than the statement of Theorem H.1.

While Theorem I.1 shows that Assumption H.1 is not vacuous, and we present a negative result in Theorem I.2 which shows exponentially decaying perturbation bounds may not hold when Assumption H.1 is not satisfied.

**Theorem I.2.** *Consider the optimal control problem where the state $x_t$ and the control input $u_t$ are both in $\mathbb{R}$. The dynamics and constraints are given by*

$$x_{t+1} = x_t + u_t, \ s.t. \ x_t \in [-1, 1], u_t \in \left[-\frac{4}{5}, \frac{4}{5}\right]$$

*for all $t$. The stage cost is given by $f_t(x_t, u_t; \xi_t) = (x_t - \xi_t)^2$, where $\xi_t = \frac{4}{5}$ if $t$ is odd, and $\xi_t = -\frac{4}{5}$ if $t$ is even. For any $p$ is even, we have*

$$\left| \psi_0^p \left(0, \xi_{0:p-1}, -\frac{2}{5}; \mathbb{I}\right)_{x_h} - \psi_0^p \left(0, \xi_{0:p-1}, -\frac{2}{5} + \epsilon; \mathbb{I}\right)_{x_h} \right| = \epsilon \tag{42}$$

*holds for any $\epsilon \in [0, \frac{2}{5(p-1)}]$ and $h \in \{1, \ldots, p\}$. For any $p$ is odd, we have*

$$\left| \psi_0^p \left(0, \xi_{0:p-1}, \frac{2}{5}; \mathbb{I}\right)_{x_h} - \psi_0^p \left(0, \xi_{0:p-1}, \frac{2}{5} + \epsilon; \mathbb{I}\right)_{x_h} \right| = \epsilon \tag{43}$$

*holds for any $\epsilon \in \left[0, \frac{2}{5p}\right]$ and $h \in \{1, \ldots, p\}$.*

Before presenting the proof of Theorem I.2, we want to add a remark about why a similar proof as Theorem I.1 cannot work here. Note that a key property we leveraged in the proof of Theorem I.1 is that any three consecutive "coupling" constraints (41) cannot activate simultaneously. This is no longer the case when $u_t$ has two sides of constraints, i.e., $u_t \in \left[-\frac{4}{5}, \frac{4}{5}\right]$. As a result, the smallest singular value of matrix $\nabla_{xy} \mathcal{L} \left(z^\dagger(\xi); \xi\right) [\mathcal{B}, \mathcal{B}]$ can be arbitrarily small (i.e., decaying w.r.t. the horizon length $p$). Thus, Assumption H.1 is not satisfied because $\underline{\sigma}_H$ cannot be set as a positive constant, and the same proof as Theorem I.1 can no longer work. We will leverage this intuition to construct a counterexample to show Theorem I.2: We construct a sequence of cost functions so that $u_t$ reaches either its lower bound $-\frac{4}{5}$ or its upper bound $\frac{4}{5}$ at every time step.

*Proof of Theorem I.2.* We first show that (42) holds by induction on $p$. Specifically, we will show that the following holds for any $q \in \mathbb{Z}_+$

$$\iota_0^{2q} \left(0, \xi_{0:2q-1}, -\frac{2}{5} + \epsilon; \mathbb{I}\right) \begin{cases} = \frac{8(q+2)}{25} + 2q\epsilon^2 & \text{if } \epsilon \in [0, \frac{2}{5(2q-1)}], \\ \geq \frac{8(q+2)}{25} + \frac{8q}{25(2q-1)^2} & \text{if } \epsilon \in (\frac{2}{5(2q-1)}, \frac{7}{5}], \\ \geq \frac{8(q+2)}{25} & \text{if } \epsilon \in [-\frac{3}{5}, 0), \end{cases} \tag{44}$$

by induction on $q$.

It is straightforward to check that (44) holds for $q = 1$. Suppose it holds for $q$. For $q + 1$, we consider the following three cases separately:

**Case 1:** $0 \leq \epsilon \leq \frac{2}{5(2q+1)}$.

Suppose $x_{2q} = -\frac{2}{5} + \epsilon'$. When $\epsilon' \in [0, \epsilon]$, we should choose $x_{2q+1} = \frac{2}{5} + \epsilon'$ to minimize the total cost. The total cost is given by

$$\frac{8(q+2)}{25} + 2q(\epsilon')^2 + \left(\frac{2}{5} - \epsilon'\right)^2 + \left(\frac{2}{5} + \epsilon\right)^2,$$

and it is minimized at $\epsilon' = \epsilon$. Thus, we achieve the total cost of $\frac{8(q+3)}{25} + 2(q+1)\epsilon^2$. When $\epsilon' > \epsilon$, note that the optimal choice of $x_{2q+1}$ is $\frac{2}{5} + \epsilon$, which is the same as when $\epsilon' = \epsilon$. By the induction assumption on $\iota_0^{2q}$, we see that the total cost incurred is lower bounded by $\frac{8(q+3)}{25} + 2(q+1)\epsilon^2$. When

$\epsilon' < 0$, we have $x_{2q+1} \leq \frac{2}{5}$. Therefore, by the induction assumption, the total cost is lower bounded by

$$\frac{8(q+2)}{25} + \frac{4}{25} + \left(\frac{2}{5} + \epsilon\right)^2 = \frac{8(q+3)}{25} + \epsilon^2 + \frac{4}{5}\epsilon \geq \frac{8(q+1)}{25} + 2(q+1)\epsilon^2.$$

Thus, we have shown that

$$\iota_0^{2q}\left(0, v_{0:2q-1}, -\frac{2}{5} + \epsilon; \mathbb{I}\right) = \frac{8(q+3)}{25} + 2(q+1)\epsilon^2, \forall \epsilon \in \left[0, \frac{2}{5(2q+1)}\right].$$

**Case 2:** $\frac{2}{5(2q+1)} < \epsilon \leq \frac{7}{5}$.

Suppose $x_{2q} = -\frac{2}{5} + \epsilon'$. When $\epsilon' \leq \frac{2}{5}$, we know that $x_{2q+1} \leq \frac{2}{5} + \epsilon' \leq \frac{4}{5}$. The total cost lower bounded by

$$\frac{8(q+2)}{25} + 2q(\epsilon')^2 + \left(\frac{2}{5} - \epsilon'\right)^2 + \left(\frac{2}{5} + \epsilon\right)^2.$$

Note that

$$\frac{8(q+2)}{25} + 2q(\epsilon')^2 + \left(\frac{2}{5} - \epsilon'\right)^2 \geq \frac{8(q+2)}{25} + 2q\left(\frac{2}{5(2q+1)}\right)^2 + \left(\frac{2}{5} - \frac{2}{5(2q+1)}\right)^2.$$

Thus the total cost is lower bounded by

$$\frac{8(q+3)}{25} + \frac{8(q+1)}{25(2q+1)^2}.$$

When $\epsilon' > \frac{2}{5}$, we see that the total cost is lower bounded by

$$\frac{8(q+2)}{25} + 2q(\epsilon')^2 + \left(\frac{2}{5} + \epsilon\right)^2$$

$$\geq \frac{8(q+2)}{25} + 2q\left(\frac{2}{5(2q+1)}\right)^2 + \left(\frac{2}{5} - \frac{2}{5(2q+1)}\right)^2 + \left(\frac{2}{5} + \frac{2}{5(2q+1)}\right)^2$$

$$= \frac{8(q+3)}{25} + \frac{8(q+1)}{25(2q+1)^2}.$$

**Case 3:** $-\frac{3}{5} \leq \epsilon < 0$.

Note that the total cost of steps 0 to $2q$ is uniformly lower bounded by $\frac{8(q+2)}{25}$ regardless of the choice of $x_{2q}$, and the total cost of steps $(2q+1)$ and $(2q+2)$ is uniformly lower bounded by $\frac{8}{25}$. Therefore, we see that

$$\iota_0^{2(q+1)}\left(0, \xi_{0:2q+1}, -\frac{2}{5} + \epsilon; \mathbb{I}\right) \geq \frac{8(q+3)}{25}.$$

Therefore, by combining the three cases, we have shown that (44) holds for all $q$ by induction.

By rolling out the optimal states that minimize the total cost, one can show the unique optimal solution is given by

$$\psi_0^p\left(0, \xi_{0:p-1}, -\frac{2}{5} + \epsilon; \mathbb{I}\right)_{x_h} = \begin{cases} \frac{2}{5} + \epsilon & \text{if } h \text{ is odd,} \\ -\frac{2}{5} + \epsilon & \text{if } h \text{ is even,} \end{cases}$$

when $\epsilon \in [0, \frac{2}{5(2q-1)}]$. This finishes the proof of (42).

For (43), suppose $p = 2q + 1$. It is straightforward to verify that (43) holds for $q = 0$. When $q \geq 1$, by (44), we know that in the optimal solution, we have $x_{2q} = -\frac{2}{5} + \epsilon$. We can further derive that the unique optimal solution is given by

$$\psi_0^p\left(0, \xi_{0:p-1}, -\frac{2}{5} + \epsilon; \mathbb{I}\right)_{x_h} = \begin{cases} \frac{2}{5} + \epsilon & \text{if } h \text{ is odd,} \\ -\frac{2}{5} + \epsilon & \text{if } h \text{ is even,} \end{cases}$$

when $\epsilon \in [0, \frac{2}{5(2q+1)}]$. This finishes the proof of (43). □