# OpenReview forum: "Bounded-Regret MPC via Perturbation Analysis: Prediction Error, Constraints, and Nonlinearity"
_NeurIPS.cc/2022/Conference — NeurIPS 2022 Accept_

### Official Review · Reviewer_g7LT · 2022-07-06

**Rating:** 8
**Confidence:** 3
**Soundness:** 4 excellent
**Presentation:** 4 excellent
**Contribution:** 4 excellent

**Summary:**

The paper formalizes a general pipeline to transform perturbation bounds into dynamic regret guarantees for model predictive control. Concretely, the pipeline has three steps (of which only one is non-immediate when employing it, see below):
1. Obtain perturbation bounds w.r.t the uncertainty parameters (1.a.) and w.r.t. the initial state (1.b.),
2. Use #1 to bound the per-step error of MPC (i.e. the distance between the actual action taken by the controller and the clairvoyant optimal action),
3. Use #2 to obtain a dynamic regret bound.

The key contribution is a "Pipeline Theorem" which takes any input from step #1 and translates it to a dynamic regret bound.

They also showcase the abilities of this pipeline by employing it to:
- extend existing dynamic regret guarantees in linear time-varying systems to handle prediction errors on costs, dynamcis and disturbances,
- provide regret bounds in nonlinear dynamics and constraints, albeit under potentially stringent assumptions.

**Questions:**

- Do we expect the resulting dynamic regret to be tight? In which cases would the analysis be prone to yielding a sub-optimal regret bound? [answered by authors]
- 1.(b) requires a bound on $y_t/v_t$, right? Is this standard? (i.e. the existing combined bounds bound this as a sub-step?) [answered by authors]

**Limitations:**

- The authors appropriately address the potential stringency of the assumptions required for the results concerning MPC with nonlinear dynamics and constraints.
- Depending on the reply to the question above regarding 'tightness', I may suggest the authors consider adding a remark regarding this point.

Minor Typo:
- line 59: 'Theorm' -> 'Theorem'

**Strengths And Weaknesses:**

Strengths:
- Successfully formalizes an analysis pipeline which I believe is quite creative/original and elegant,
- The paper is presented very well, with a lot of intuitions and very clear statements,
- The pipeline is also employed to obtain new results, which is evidence of the pipeline's usefulness.

Weaknesses:
- While the writing is very clear, the notation summary is deferred to the appendix which may compromise readability a bit (e.g. maybe could add a reference to the Appendix in the preliminaries before the notations start being used so people can click and be redirected?),
- No formal 'Conclusion' section.

---

> ### Author Response · Authors · 2022-08-02
> **Reply to Reviewer g7LT**
>
> Thanks for your comment and please find the response to your questions below:
>
> 1. *Response to the tightness of our regret bounds:*
>
> Yes, we expect the exponentially decaying form of the coefficients before prediction errors and the dependence on prediction horizon is tight. However, the decay factor $\lambda$ might not match the optimal decay factor. Similar situations regarding upper and lower bounds have been found in a special case of the LTI system in [39].
>
> 2. *Response to the bound on $y_t$ and $v_t$ in 1.(b):*
>
> Yes. We believe this is standard. In [35], although the perturbation bound in Theorem 3.2 only considers $y_t$, the proof also requires $v_t$ to satisfy the same kind of perturbation bound (see the equations between (21) and (22)). Therefore, we believe it is necessary to require the perturbation bound to hold for both $y_t$ (predictive state) and $v_t$ (predictive action) to show the dynamic regret bound of MPC in the general case.

---

> > ### Comment · Reviewer_g7LT · 2022-08-08
> > **Thanks for the reply!**
> >
> > Thank you for the replies! I maintain my score.

---

### Official Review · Reviewer_UZES · 2022-07-06

**Rating:** 5
**Confidence:** 3
**Soundness:** 3 good
**Presentation:** 3 good
**Contribution:** 2 fair

**Summary:**

The authors provide an analysis of dynamic regret in model predictive control by identifying how perturbation bounds for finite-time optimal control problems affect the closed-loop dynamic regret in MPC. The authors then instantiate their pipeline theorem on two examples with linear time-varying (LTV) dynamics, and one general nonlinear dynamics with constraints example.

**Questions:**

Line 39: Can the authors elaborate on what this “counter-intuitively large re-planning window” refers to?

Line 127: Is it assumed that the true parameters $\xi_t^* \in \Xi_t$? I presume so, otherwise the prediction error cannot be driven arbitrarily small.

Algorithm 1, Line 4: The notation $\psi_t^{t’}(\cdot)_{v_t}$ is confusing. Consider something like $\psi_t^{t’}(\cdot)[0]$ instead.

Definition 3.1: What happens when the solutions $\psi_t^T$ are not unique? Does the definition just rely on picking an arbitrary optimal solution?

Lines 181-182: Regarding “the per-step error converges to zero as the prediction horizon $k$ increases and the quality of predictions improves (i.e., $\|\xi_{t:t+k|t} - \xi^*_{t:t+k}\| \to 0$)”: it seems that as $k$ grows, $\|\xi_{t:t+k|t} - \xi^*_{t:t+k}\|$ should grow as well (and not tend to zero), since the predictor is being asked to predict over a longer horizon $[t:t+k]$ using only the information at times $[0:t]$.

Equation (4): What does the $y_t/v_t$ subscript mean?

Property 3.1: I think $\xi_{t+k}, \xi’_{t+k} \in \mathcal{B}(x_{t+k}^*, R)$ is a typo. Furthermore, on Line 218, “perturbation bound (4) holds for any $z,z’ \in \mathcal{B}(x_t^*, R)$”, is this for the true parameters $\xi_{t}^*$?


**Limitations:**

The authors acknowledge (in Lines 344-351) that their analysis in Theorem 5.1 (a) does not tackle the question of recursive feasibility for MPC, and (b) the algorithm is not actually practical since it requires setting the terminal state $\bar{y}(\zeta_{t+k|t}) \in \mathcal{B}(x_{t+k}^*, R)$, but the optimal trajectory $x_t^*$ is not known.

**Strengths And Weaknesses:**

**EDIT:** After reading the author's responses, I am raising my score to a borderline accept.

**Strengths:**

The authors tackle a very challenging problem of controlling the regret of MPC in constrained nonlinear settings.

**Weaknesses:**

It is unclear how much the presented results advance our understanding of MPC. Specifically:

- Regarding Section 4.1, Theorem 4.1: In Line 298, it is noted that when there are no prediction errors, this result recovers the existing result in [35]. On the other hand, when there are prediction errors, the term $\sum_{\tau} \lambda_i^\tau P(\tau)$ needs to be $o(T)$ for the dynamic regret to be $o(T)$. It is not clear when this can happen, given that the prediction error is over the disturbance term $w_t(\xi_t^*)$ in Eq. (9). The two most common cases of modeling disturbances are (a) iid stochastic terms, and (b) bounded oblivious/adversarial terms. In either case, it seems that $P(\tau) = \Omega(T)$, since past observations give no information about future observations.

- Regarding Section 4.2, Theorem 4.2: In addition to the previous paragraph, Assumption G.1 on the unknown dynamics $\{ (A_t(\cdot), B_t(\cdot)) \}$ seems quite restrictive, by asking for uniform controllability over all uncertainty parameters $\xi_t$. In practice, this may substantially limit the uncertainty set $\Xi_t$ so that the parameters $\{ (A_t, B_t) \}$ are effectively known.

- Regarding Section 5, Theorem 5.1: Assumption H.1 is stated in purely technical terms, and no examples of even toy systems are given that satisfy all the assumptions. One very practical setting which seems like it cannot be handled by the theory is when one of the constraints is active, i.e., $s_t(x_t, u_t, \zeta_t^*) = 0$. Intuitively, it seems like Property 3.1 cannot hold in that case, since small perturbations in either the parameters $\zeta_t$ or the initial condition $x_0$ may cause the MPC problem to be either infeasible or jump to a very different solution. Furthermore, it is not clear the condition $H_3 \sum_{\tau=0}^{k-1} \lambda_3^\tau \rho_{t,\tau} + 2 R H_3 \lambda_3^k \leq \frac{(1-\lambda_3)^2 R}{H_3^2 L_g}$ can be practically satisfied; as $\lambda_3 \to 1$, this condition becomes more stringent, requiring increasingly accurate prediction errors $\rho_{t,\tau}$.

I am open to changing my score if the authors can construct some specific systems which satisfy all the technical assumptions, specifically in Theorem 4.2 and Theorem 5.1.

---

> ### Author Response · Authors · 2022-08-02
> **Reply to Reviewer UZES (Part 1/2)**
>
> Thanks for your comment and please find our response below:
>
> 1. *Response regarding achieving o(T) dynamic regret:*
>
> While it is possible to achieve sublinear regret against a static offline optimal [41], it is impossible to achieve such order of regret bounds against the dynamic offline optimal in the worst case. One can show that the lower bound in the stochastic i.i.d. case is $\Omega(T)$ (see [46]).
>
> Our goal is not to break the $\Omega(T)$ lower bound in the worst case. Instead, we provide a refined characterization of the dynamic regret bound, which contains a weighted sum of the $\tau$ step prediction errors, and the weight decays exponentially in $\tau$. Such a characterization has many interesting implications. In some cases, the prediction error for disturbances and dynamical systems can be very close to zero for a certain time horizon (e.g., trajectory tracking, smoothed regression, and frequency regulation. See the examples discussed in [35] and [38].) and in such cases, the dynamic regret can be very small.
>
> 2. *Response to the uniform controllability assumption in Theorem 4.2:*
>
> If the dynamics is not assumed to be uniformly controllable for all possible uncertainty parameters, it is possible that the control input solved by MPC is generated by an instance $\xi$ such that the corresponding $M_t$ is not or nearly not controllable, which incurs a large stage cost for MPC. The uniform controllability assumption is reasonable in many applications, e.g.,
> * First, consider the example of controlling a linearized inverted pendulum dynamics with unknown mass. It is clear that as long as the mass is within some range, the uniform controllability condition will hold.
> * Second, consider the LTV frequency regulation with unknown inertia in [18]. In equation (9) of [18], when the equivalent rotational inertia $m_{q(t)}$ is within a range $[a, b]$ where $0 < a < b$, one can show that the LTV system considered in [18] is uniformly controllable.
>
> 3. *Response regarding the assumptions and conditions of Theorem 5.1:*
>
> We found good and bad instances can arise even in very simple dynamical systems. For example, consider the 1-dimensional dynamical system given by $x_{t+1} = x_t + u_t$. In the first case, we consider a simple example where the constraints are $u_t \ge - 0.8$ (one-sided) and $-1 \le x_t \le 1$. We can show Assumption H.1 holds, thus the exponential decay perturbation bounds hold. However, if we change the constraints on $u_t$ to be $0.8 \ge u_t \ge - 0.8$ (two-sided), one can construct a counterexample where the exponential decay perturbation bounds do not hold. We give the detailed proofs of these two examples in Appendix I in the revised supplementary material. The first positive case shows that Assumption H.1 is not vacuous, and the second negative case shows exponentially decaying perturbation bounds may not hold when Assumption H.1 is not satisfied.
>
> We also want to emphasize that MPC with constraints are very hard in general [Mayne et al., 2000]. Despite the difficulty, our work makes a contribution towards understanding MPC’s performance in this case. Specifically, our results provide a reduction from the MPC’s dynamic regret bound to the perturbation bound in FTOCP problems, which is a much better understood problem in optimization literature [37,50].
>
> While the condition on $\rho$ and $k$ in Theorem 5.1 may appear strong, note that there is an exponential decay coefficient before the prediction error $\rho_{t,\tau}$. Therefore, the key to satisfy this condition is to ensure the first few steps of prediction are sufficiently accurate. This is reasonable to achieve in many applications, e.g. in power system frequency regulation, and in trajectory tracking.
>
> [Mayne et al., 2000] Mayne, David Q., James B. Rawlings, Christopher V. Rao, and Pierre OM Scokaert. "Constrained model predictive control: Stability and optimality." Automatica 36, no. 6 (2000): 789-814.

---

> > ### Comment · Reviewer_UZES · 2022-08-06
> > **thanks for the response**
> >
> > Thanks for the detailed response. In my original review, I expressed concerns about the following issues:
> >
> > 1. The issue of dynamic regret possibly scaling as $\Omega(T)$.
> > 2. The uniform controllability assumptions of Theorem 4.2.
> > 3. The opaqueness about the assumptions of Theorem 5.1.
> >
> > Regarding (3) first, I took a look at the new results included in the supplementary material, and they are nice.
> >
> > Regarding (2), could the authors also explicitly instantiate one of the uniform controllability examples quoted in the rebuttal? Specific examples bring a lot of clarity to the general theorems.
> >
> > Regarding (1), thanks for pointing to the $\Omega(T)$ lower bound. The authors should consider including the discussion about the lower bound, and the point of the dynamic regret results discussed in the rebuttal (if they are not already included in the manuscript).
> >
> > Based on the response, I am raising my score to a borderline accept. I think the paper is a nice technical contribution. What prevents this paper from being a strong accept is that, it was not until the reading the authors responses to both other reviewer's rebuttals and my own that I started to appreciate the technical contributions of this work, and the non-triviality of the results. As pointed out by reviewer MMqe, readers who are not intimately familiar with the finite-time MPC literature may have a hard time understanding the significance of the results.

---

> > > ### Author Response · Authors · 2022-08-08
> > > **Thanks for your comment and the follow-up question**
> > >
> > > Thanks for your comment and the follow-up question. We have updated the supplementary material and added more details about the two examples of uniform controllability at the end of Appendix G. We will add more discussion about the technical challenges and contributions of our work in the final revision.

---

> ### Author Response · Authors · 2022-08-02
> **Reply to Reviewer UZES (Part 2/2)**
>
> 4. *Response regarding "the counter-intuitively large re-planning window":*
>
> Say, e.g., the RHC controller used in [48], which introduces a “lag $L$” between consecutive horizons. This is counter-intuitive since, if we obtain information of better precision at each time step, using the newest information retrieved in each step should always be better than using the information retrieved $L$ steps ago.
>
> 5. *Response to the uniqueness the solutions $\psi_t^T$:*
>
> Yes, the solutions $\psi_t^T$ in the definition relies on picking certain optimal solutions (if there are more than one) that satisfy the assumptions (Property 3.1) to apply the pipeline theorem.
>
> 6. *Response regarding the $y_t/v_t$ subscript:*
>
> We use ${\psi_{t}^{t’}(\cdot)}$ with subscript $y_\tau$ to denote the state at time $\tau$, and we use ${\psi_{t}^{t’}(\cdot)}$ with subscript $v_\tau$ to denote the control input at time $\tau$, both referring to the FTOCP solution used by the predictive controller. See Definition 2.3, where the optimization variables are $y_{t:t’}$ and $v_{t:t’-1}$.
>
> 7. *Response regarding Property 3.1:*
>
> We clarify that, in the applications of our Pipeline Theorem (see Section 4 and 5), $\xi_{t+k}$ is always in the same space as the state $x_{t+k}$, so it is not a prediction of the uncertainty parameter as $\xi_{t:t+k-1}$. We have already defined this in Definition 2.3 (see line 148). We define it in this way so that in the algorithm pseudo code, the first $T-k$ steps (which use indicator terminal costs) can be unified with the last $k$ steps (which use the prediction of true terminal cost $F_T$).
> We understand this may be confusing, and in revision, we will change the notation to: $\zeta_{t+k}, \zeta_{t+k}' \in \mathcal{B}(x_{t+k}^*, R).$
> For line 218, we mean the perturbation bound holds for true parameters $\xi_t^*$. We will clarify this in the revision.

---

### Official Review · Reviewer_MMqe · 2022-07-16

**Rating:** 6
**Confidence:** 3
**Soundness:** 3 good
**Presentation:** 3 good
**Contribution:** 3 good

**Summary:**

This paper proposes a general analysis pipeline to bound the dynamic regret of MPC problem. The proposed pipeline has 3 steps: (1) obtain the perturbation bounds  which relies on the specific form of MPC formulation, (2) bound the per-step error using the perturbation bounds, (3) bound the dynamic regret by perturbation bounds. The proposed pipeline is then applied to generalize existing regret bounds on MPC.

**Questions:**

For readers who are not closely following the recent advances on finite-time analysis of MPC, like me, can authors clarify the technical contributions of the paper with respect to the existing works. For example,

(1) How different are the three-steps analysis in the proposed pipeline with the dynamic regret analysis for MPC in the literatures? The authors just summarized and abstracted the existing analysis techniques or come up with a new set of techniques?

(2) What are the technical challenges for MPC problems considered in section 4 and 5 such that no existing works solve them? For example, can the technique in [35] be extended to the problem (9) by adding the prediction error and using some common inequalities?

(3) How valuable is Theorem 5.1 given its slighly strong assumption and the fact that the FTOCP can not be guaranteed to be solved to the globally optimal solution for constrained nonlinear systems.

**Strengths And Weaknesses:**

Strengths:

(1) The paper is well-written and clear.

(2) A novel analysis pipeline to bound the dynamic regret of MPC problem. This analysis pipeline not only unifies the previous results but also generalize to the new settings which have not been well-studied.

Weaknesses:

(1) Theorem 5.1 relies on a strong assumption and restricts its implications.

(2) The technical challenge of the proposed pipeline and the new settings considered in Section 4 and 5 are not clear. Without these, it is hard to evaluate the technical contributions of the paper.

---

> ### Author Response · Authors · 2022-08-02
> **Reply to Reviewer MMqe**
>
> Thanks for your comment and please find our response below:
>
> 1. *Response to the novelty of our analysis and the difference with previous work (Question 1 and 2):*
>
> While [35] proposed the perturbation analysis approach, our current paper considers a much more general setting compared with [35] that includes prediction errors (even on dynamical matrices), nonlinear dynamics, and constraints. This generalization is much more challenging compared with the setting and proof techniques in [35]. For example, to account for the prediction error, we need to first bound the per-step error of MPC, and then study the cumulative impact of per-step errors on the distance between $x_t$ and $x_t^*$. Compared with [35] which directly bound the norm of $x_t$, a significant advantage of our pipeline approach is that, when perturbation bounds only hold locally near OPT (as in Section 5) rather than globally (as in [35]), the Step 2 and Step 3 in the pipeline can work together to guarantee the analysis can proceed inductively: On the one hand, $e_t$ needs to be small so that $x_t$ can stay in the neighborhood where the perturbation bounds holds. On the other hand, it is important to guarantee the perturbation bounds hold so that the next per-step error $e_{t+1}$ is small.
>
> In addition, we also make other technical contributions by showing a new perturbation bound that allows the perturbations on LTV dynamical matrices in the setting of Section 4.2. This is beyond the perturbation bound results shown by [35], which can only be used to study the prediction errors on disturbances (Section 4.1).
>
> 2. *Response regarding the assumptions of Theorem 5.1 and finding the global optimal solution (Question 3):*
>
> It is well-known that bounding MPC’s dynamic regret in nonlinear systems with constraints is very challenging. Our contribution in Theorem 5.1 lies in the reduction from the problem of MPC dynamic regret to the perturbation analysis of FTOCP for the corresponding system, which we believe is more tractable and has been studied heavily in the optimization literature [37, 48, 49 ,50].How to solve the FTOCP and how to guarantee the optimality of the solution is beyond the scope of this paper. A separate line of work has studied the impact of inexact solutions on MPC performance (e.g., [Zeilinger et al., 2011]) and we will add more references and discussions in revision.
>
> [Zeilinger et al., 2011] Zeilinger, Melanie Nicole, Colin Neil Jones, and Manfred Morari. "Real-time suboptimal model predictive control using a combination of explicit MPC and online optimization." IEEE Transactions on Automatic Control 56, no. 7 (2011): 1524-1534.

---

### Official Review · Reviewer_qCVx · 2022-07-18

**Rating:** 5
**Confidence:** 3
**Soundness:** 3 good
**Presentation:** 3 good
**Contribution:** 2 fair

**Summary:**

This paper studies how to bound the dynamic regret for MPC. The analysis is built upon a new perturbation bound for finite-horizon optimal control.  Per-step error of MPC is bounded for regret analysis. The proposed analysis covers a variety of settings. Some existing results for the LTV case has been recovered and strengthened. New results have also been obtained for the nonlinear case.

**Questions:**

My main questions are:

1)  Is there any way to justify the assumptions for practical problems? How realistic is it to assume the stability in nonlinear setting?

 2) What new insights for algorithm design are brought by the proposed regret analysis?

**Limitations:**

Yes, the authors have discussed the limitations in the end of Section 5.

**Strengths And Weaknesses:**

Strengths:
1. The perturbation bound for finite-horizon optimal control looks interesting and original.
2. Several different settings are unified using the proposed approach.
3. New results have been obtained for both LTV and nonlinear cases.

Weaknesses:
1. The presentation is not that clear. Section 2 may be revised with the focus on MPC. The connections between Problem (1) and MPC need more explanations.
2. The assumptions are very strong: stability is assumed in the first place. That seems quite unreasonable.
3. It seems that the regret analysis does not lead to any end-to-end algorithms for practical MPC. Any insights can be obtained for algorithm design?

---

> ### Author Response · Authors · 2022-08-02
> **Reply to Reviewer qCVx**
>
> Thanks for your comment and please find the response to your questions below.
>
> 1. *Response to the assumption on stability:*
>
> We clarify that we do not assume the MPC controller to be stable. We only assume the offline optimal solution to be stable - more specifically, we assume the optimal trajectory $x_t^*$ lies in a bounded ball.
>
> In fact, to prove our regret bound, we show that the MPC controller is stable (see Lemma 3.2) under the assumptions of the Pipeline Theorem 3.3.
>
> 2. *Response to new insights for algorithm design:*
>
> Previous works (e.g. [33] and [35]) demonstrate the benefits of having a longer prediction horizon, but they are based on the assumption that all predictions are exact. In contrast, our results show that this may not be true when predictions are inexact. Specifically, the regret depends on a weighted sum of the $\tau$-step prediction errors, with the weight decaying exponentially w.r.t. $\tau$. This shows that depending on specific prediction error patterns, choosing an appropriate prediction horizon can be important. Increasing the prediction horizon but with very inaccurate predictions can actually hurt the dynamic regret.
>
> 3. *Response regarding the presentation of Section 2:*
>
> Our paper studies how to solve the optimal control problem (1) online with access to noisy future predictions. MPC is a widely-used approach to solve this problem by calculating sub-trajectory confined to a finite horizon [1]. Correspondingly, in Section 2 we first introduce the problem (1), define the predictive online control setup, and then introduce the MPC algorithm. In revision, we will add more explanations on the connections between (1) and MPC and streamline the presentation.

---

> > ### Comment · Reviewer_qCVx · 2022-08-08
> > **thanks for the response**
> >
> > Thanks for the response. I am still not very convinced by the insight brought by the proposed analysis to algorithm design. The authors' main insight is "increasing the prediction horizon but with very inaccurate predictions can actually hurt the dynamic regret." Maybe I am missing something here. This seems to be common sense to me. I am not sure whether we need to do this type of regret analysis to realize that inaccurate predictions can hurt MPC's performance. It seems obvious that we want to use robust MPC or other versions of robust control techniques to address the inaccurate predictions in system parameters, right? It seems a little bit weird to me that robust MPC is not even discussed in this paper. I mean, for systems with model errors, that seems to be the first thing coming to mind, right?
> > The key design implication of the theory in this paper seems to be that the regret will be small if the prediction errors are sufficiently small. Again, this seems to be common sense. Can the theory of this paper lead to some "optimal" window length in theory? Can the theory of this paper be extended to robust MPC and demonstrate that robust MPC improves over MPC with inaccurate predictions? The theory will be much stronger if the analysis can bring some quantitative characterization of the improvement of robust MPC over MPC. Right now, I am very struggling to see any useful design insights other than some common knowledge. Providing a regret bound is theoretically interesting. However, explaining how useful these bounds are is equally important. I also agree with other reviewers' opinion that the assumptions in Theorem 5.1 are very strong. This makes the application of such a result in practical nonlinear control even more difficult.

---

> > > ### Author Response · Authors · 2022-08-08
> > > **Thanks for the comment and follow-up questions**
> > >
> > > Thank you for your comments and follow-up questions. Please find the response to your questions below:
> > >
> > > > 1. *The authors' main insight is "increasing the prediction horizon but with very inaccurate predictions can actually hurt the dynamic regret." Maybe I am missing something here. This seems to be common sense to me. I am not sure whether we need to do this type of regret analysis to realize that inaccurate predictions can hurt MPC's performance.*
> > >
> > > While this intuition is true generically, the theory gives us a more precise understanding of when and why. Consider the dynamic bound in Theorem 3.3. If the prediction error $P(\tau)$ increases exponentially with $\tau$, but the exponential rate is smaller than the inverse of the decay rate of $q_1$ and $q_2$, the regret will not become worse (up to a constant factor) when the prediction horizon $k$ increases. In contrast, when the error rate increases exponentially with rate faster than the inverse of the perturbation bounds’ decay rate, the inclusion of such prediction will cause much worse regret. This insight highly relies on the specific form of our regret bound and we do not believe it is trivial or known before.
> > >
> > > >2. *It seems obvious that we want to use robust MPC or other versions of robust control techniques to address the inaccurate predictions in system parameters, right? It seems a little bit weird to me that robust MPC is not even discussed in this paper. I mean, for systems with model errors, that seems to be the first thing coming to mind, right?*
> > >
> > > We want to emphasize that, while robust MPC is an important class of algorithms to handle model uncertainties, the focus of our paper is to understand how prediction errors impact the performance of standard MPC. We believe the problem we study is important because standard MPC is widely used in practice and existing theoretical results have significant limitations [33, 34, 35]. As we will discuss in Point 5, extending our pipeline framework to bound the regret of robust MPC is an interesting future direction.
> > >
> > > >3. *The key design implication of the theory in this paper seems to be that the regret will be small if the prediction errors are sufficiently small. Again, this seems to be common sense.*
> > >
> > > As we discussed in Point 1, the prediction error does not need to be small if it is about the far future. Specifically, it can even increase exponentially as long as the rate is smaller than the inverse of the decay rate of the perturbation bound 3.1 (a).
> > >
> > > >4. *Can the theory of this paper lead to some "optimal" window length in theory?*
> > >
> > > Our dynamic regret bound in Theorem 3.3 provides some insight on how to choose the window length: As the prediction window size increases from $k$ to $k + 1$, the last term $(q_1(k)^2 + q_2(k)^2) T$ in $E$ will decrease to $(q_1(k+1)^2 + q_2(k+1)^2) T$, but an extra term $(q_1(k)^2 + q_2(k)^2) P(k)$ will be added to the first term in $E$. Thus, whether the dynamic regret improves depends on both the total prediction error after $k$ steps $P(k)$ and the specific form of perturbation bounds.
> > >
> > > >5. *Can the theory of this paper be extended to robust MPC and demonstrate that robust MPC improves over MPC with inaccurate predictions? The theory will be much stronger if the analysis can bring some quantitative characterization of the improvement of robust MPC over MPC.*
> > >
> > > Our analysis partially holds for robust MPC: Specifically, Step 3 (Lemma 3.2) about the accumulation of per-step errors still applies. The remaining challenge is at Step 2. We need to bound the per-step error incurred by Robust MPC, which requires a new type of perturbation bound about the robust optimization problem. To the best of our knowledge, such bounds are open. To extend our results to Robust MPC, we are currently exploring different forms of robust MPC (e.g., [Bujarbaruah et al., 2021]). We will add a discussion about this topic in the final revision.
> > >
> > > [Bujarbaruah et al., 2021] Bujarbaruah, Monimoy, Ugo Rosolia, Yvonne R. Stürz, and Francesco Borrelli. "A simple robust MPC for linear systems with parametric and additive uncertainty." In 2021 American Control Conference (ACC), pp. 2108-2113. IEEE, 2021.
> > >
> > > >6. *I also agree with other reviewers' opinion that the assumptions in Theorem 5.1 are very strong. This makes the application of such a result in practical nonlinear control even more difficult.*
> > >
> > > We want to emphasize that MPC with constraints and nonlinear dynamics are very hard in general. Despite the difficulty, our work makes a contribution towards understanding MPC’s performance in this case. Specifically, our results provide a reduction from the MPC’s dynamic regret bound to the perturbation bound in FTOCP problems, which is a much better understood problem in optimization literature [37,50]. We also use two examples in a simple dynamical system to justify Assumption H.1 for Theorem 5.1 (see Appendix I in the supplementary material).

---

### Meta-Review · Area_Chair_Uu9u · 2022-08-31

**Recommendation:** Accept
**Confidence:** Less certain

**Metareview:**

The paper studies MPC and proposes a general analytic framework to bound dynamic regret. The approach is to achieve a perturbation bound for the MPC problem per step and then obtain a regret bound.

Overall the paper saw a lot of discussion, with the reviewers initially questioning the non-triviality of the results, but those questions were sufficiently addressed. Finally, the reviewers have liked the theoretical contributions of the paper, however the practical insight or impact has been particularly questioned. The lack of results on Robust MPC has been a talking from the point of view of whats used in practice, under these assumptions. The paper from the reviewer's assessments lay on the borderline, but the theoretical contributions are strong and that leads to me to a recommendation of marginal accept.

However, this recommendation comes with a strong plea to the authors regarding improving the manuscript, especially improving discussion and presentation of recent results on regret analysis of MPC. This is important so that readers can fully comprehend and understand the contribution of the paper.

**Award:**

No

---

### Decision · Program_Chairs · 2022-09-14

Accept